# NOVA3R: NON-PIXEL-ALIGNED VISUAL TRANSFORMER FOR AMODAL 3D RECONSTRUCTION

**Weirong Chen**[1,2]**, Chuanxia Zheng**[3,4*]**, Ganlin Zhang**[1,2]**, Andrea Vedaldi**[3]**, Daniel Cremers**[1,2]
[1]Technical University of Munich     [2]Munich Center for Machine Learning
[3]University of Oxford     [4]Nanyang Technological University

## ABSTRACT

We present NOVA3R, an effective approach for non-pixel-aligned 3D reconstruction from a set of unposed images in a feed-forward manner. Unlike pixel-aligned methods that tie geometry to per-ray predictions, our formulation learns a global, view-agnostic scene representation that decouples reconstruction from pixel alignment. This addresses two key limitations in pixel-aligned 3D: (1) it recovers both visible and invisible points with a complete scene representation, and (2) it produces physically plausible geometry with fewer duplicated structures in overlapping regions. To achieve this, we introduce a scene-token mechanism that aggregates information across unposed images and a diffusion-based 3D decoder that reconstructs complete, non–pixel-aligned point clouds. Extensive experiments on both scene-level and object-level datasets demonstrate that NOVA3R outperforms state-of-the-art methods in terms of reconstruction accuracy and completeness. Our project page is available at: https://wrchen530.github.io/nova3r.

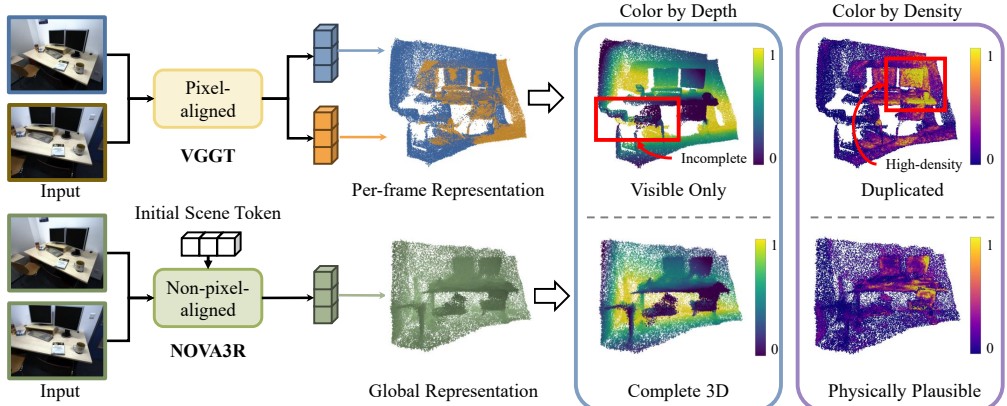

Figure 1: **NOVA3R** enables non–pixel-aligned reconstruction by learning a global scene representation from unposed images. Compared to pixel-aligned methods, NOVA3R recovers both visible and occluded regions and produces more physically plausible geometry with fewer duplicated structures.

## 1 INTRODUCTION

We consider the problem of *non–pixel-aligned* 3D reconstruction from one or more unposed images, in a feed-forward manner. This is a challenging task, as the model must infer a global, view-agnostic representation of the scene without relying on per-ray supervision. This formulation avoids the limitations of pixel-aligned methods, which reconstruct only visible surfaces and often produce redundant geometry in overlapping regions. It therefore enables more complete and physically plausible 3D reconstruction, capturing both visible and occluded structures in a consistent manner.

Recent work in 3D reconstruction has largely focused on the *pixel-aligned* formulation, where geometry is predicted in the form of depth maps, point maps, or radiance fields tied to the image plane.

---

*Project Lead

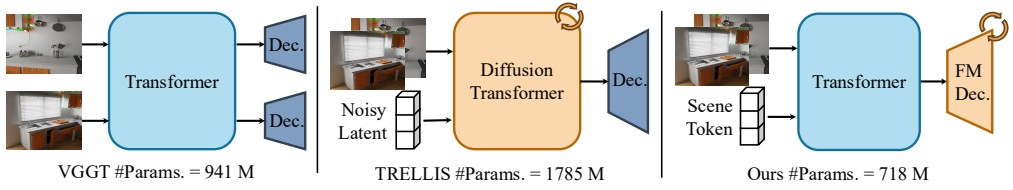

Figure 2: **Comparison of different reconstruction paradigms.** Our non-pixel-aligned approach combines feed-forward efficiency with a global, view-agnostic scene representation, removing the reliance on pixel-level supervision. NOVA3R provides a unified solution for various reconstruction tasks, achieving multi-view consistency and geometrically faithful results.

DUSt3R (Wang et al., 2024a) pioneers this paradigm of dense, pixel-aligned 3D reconstruction from unposed image collections, achieving impressive results in reconstructing the visible regions of a scene. Building on this, follow-up works (Tang et al., 2025b; Wang et al., 2025b; Yang et al., 2025; Zhang et al., 2025b; Wang et al., 2025a) extend DUSt3R from image pairs to multi-view settings, enabling feed-forward 3D geometry reconstruction from larger image sets. However, the pixel-aligned formulation remains tied to per-ray prediction, which restricts reconstruction to visible regions and yields *incomplete* geometry and *overlapping point layers* in areas visible to multiple cameras.

Another line of work explores latent 3D generation, which learns a *global representation* in a compact latent space and decodes it into voxels or meshes (Vahdat et al., 2022; Zhang et al., 2023; 2024b; Ren et al., 2024; Xiang et al., 2025a; Tochilkin et al., 2024; Team, 2024; 2025; Li et al., 2025b). While this global formulation can plausibly complete occluded regions beyond the input views, most approaches remain confined to the *object level*. They assume canonical space and require high-quality mesh supervision, which makes these methods struggle with complex, cluttered scenes. For *scene*-level reconstruction, some methods (Chen et al., 2024; Liu et al., 2024; Gao et al., 2024; Szymanowicz et al., 2025) inpaint unseen regions by synthesizing novel views with pre-trained diffusion models and then post-process to recover geometry. However, such pipelines do not guarantee physically meaningful point clouds.

To overcome these limitations, we introduce the Non-pixel-aligned Visual Transformer (NOVA3R) (see Figure 1). First, we address the challenge of non-pixel-aligned supervision by leveraging a diffusion-based 3D autoencoder. It first compresses complete point clouds into compact latent tokens, and then decodes them back into the original space, supervised with a flow-matching loss that resolves matching ambiguities in unordered point sets. Recent works on 3D autoencoders (Zhang et al., 2023; Xiang et al., 2025a; Team, 2024; Li et al., 2025b) have demonstrated the effectiveness of latent representations, but they are primarily designed for object reconstruction, assuming high-quality meshes for supervision. In contrast, our formulation targets scene-level reconstruction and requires only point clouds derived from meshes or depth maps for supervision, enabling it to capture priors of complete 3D scenes and produce physically coherent geometry without duplicated points.

Second, we tackle the problem of mapping unposed images to a global scene representation. Training such a model directly would require massive amounts of complete scene data and computational resources. To improve generalization, our model is built on a pre-trained image encoder from VGGT (Wang et al., 2025a), augmenting it with learnable scene tokens that aggregate information from arbitrary numbers of views and map them into the latent space of our point decoder. This design enables NOVA3R to support both monocular and multi-view reconstruction, without being restricted to a fixed number of inputs. Despite being trained on relatively small datasets, our model generalizes well to unseen scenes, achieving complete and physically plausible reconstructions.

In summary, our main contributions are as follows: (i) We introduce a unified non-pixel-aligned reconstruction pipeline with minimal assumptions, applicable to both object-level and scene-level complete reconstruction tasks. (ii) We address key limitations of pixel-aligned methods, which often produce incomplete point clouds, duplicated geometry, and 3D inconsistencies in overlapping regions. By contrast, our non-pixel-aligned formulation naturally yields complete and evenly distributed geometry. (iii) We integrate a feed-forward transformer architecture with a lightweight flow-matching decoder, effectively bridging the gap between pixel-aligned reconstruction and latent 3D generation, combining feed-forward efficiency with strong 3D modeling capability (see Figure 2).

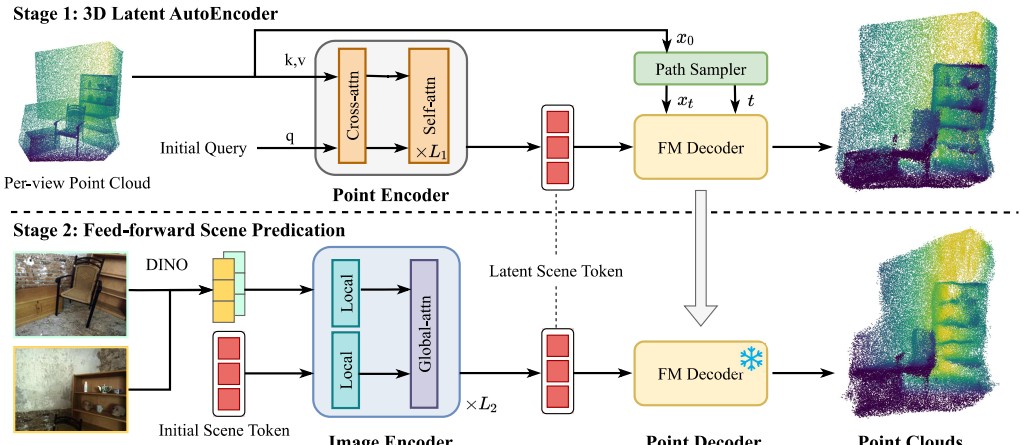

Figure 3: **Overview of NOVA3R. Stage 1:** a 3D point autoencoder encodes complete point clouds into latent scene tokens and decodes them with a flow-matching (FM) decoder. **Stage 2:** an image encoder with learnable scene tokens integrates multi-view information into a unified scene latent space, supervised by the FM loss with the Stage-1 decoder frozen. During **inference**, only the second stage pipeline is used to produce a complete, non–pixel-aligned point cloud.

## 2 RELATED WORK

**Feed-Forward 3D Reconstruction.** Unlike *per-scene* optimization methods (Mildenhall et al., 2020; Kerbl et al., 2023) that iteratively refine a 3D representation for each individual scene, *feed-forward* 3D reconstruction approaches aim to generalize across scenes by predicting 3D geometry directly from a set of input images in a single pass of a neural network. Early approaches typically focus on predicting geometric representations, such as depth maps (Eigen & Fergus, 2015), meshes (Wang et al., 2018), point clouds (Fan et al., 2017), or voxel grids (Choy et al., 2016), and are trained on relatively small-scale datasets (Nathan Silberman & Fergus, 2012; Chang et al., 2015). As a result, these models struggled to capture fine-grained visual appearance and exhibited limited generalization to unseen scenes.

More recently, DUSt3R (Wang et al., 2024a) and MASt3R (Leroy et al., 2024) directly regress dense, pixel-aligned point maps from unposed image collections. These approaches mark a significant step toward generalizable, pose-free 3D reconstruction. Building on this paradigm, many recent works (Tang et al., 2025b; Wang et al., 2025b; Yang et al., 2025; Zhang et al., 2025b; Wang et al., 2025a) extend it from image pairs to multi-view settings, enabling feed-forward 3D geometry reconstruction from sets of uncalibrated images. However, these pixel-aligned methods produce incomplete geometry and duplicated points in overlapping regions. In contrast, our approach outputs a unified and *complete* 3D reconstruction that integrates both *visible* and *occluded* regions.

**Complete 3D Reconstruction.** To achieve a complete 3D reconstruction, existing approaches typically follow two main paradigms. One line of work (Vahdat et al., 2022; Zhang et al., 2023; Zhao et al., 2023; Zhang et al., 2024b; Ren et al., 2024; Xiang et al., 2025a; Tochilkin et al., 2024; Team, 2024; 2025; Li et al., 2025b) leverages compact latent spaces (Rombach et al., 2022) or large-scale networks (Hong et al., 2024; Zhang et al., 2024a; Tang et al., 2025a) for generating complete 3D assets. While effective, these approaches primarily target individual *object* reconstruction and fall short in modeling complex, cluttered scenes. The other paradigm fine-tunes large-scale pre-trained diffusion models (Rombach et al., 2022; Blattmann et al., 2023). For *objects*, a notable example is Zero-1-to-3 (Liu et al., 2023b), which conditions on camera pose for high-quality 360-degree novel view rendering by training on a huge dataset, Objaverse (Deitke et al., 2023). This is followed by a large group of successors (Long et al., 2024; Shi et al., 2024; Han et al., 2024; Liu et al., 2023a; Li et al., 2024; Zheng & Vedaldi, 2024; Ye et al., 2024; Voleti et al., 2024). For *scenes*, several recent approaches aim to achieve complete 3D geometry by leveraging controlled camera trajectories (Wang et al., 2024b; Sargent et al., 2024; Wu et al., 2024; Gao et al., 2024; Wallingford et al., 2024; Zhou et al., 2025) or introducing auxiliary conditioning signals (Liu et al., 2024; Yu et al., 2025b; Chen et al., 2024; Yu et al., 2025a). However, these methods do not explicitly

reconstruct the complete underlying 3D geometry. More recently, WVD (Zhang et al., 2025a) and Bolt3D (Szymanowicz et al., 2025) propose a hybrid RGB+point map representation to combine geometry and appearance for 3D reconstruction; however, they still require known camera poses for novel RGB+point map rendering. We address *pose-free* 3D reconstruction from unconstrained images, and provide a complete 3D representation. More closely related to our work, Amodal3R (Wu et al., 2025) introduces amodal 3D reconstruction to reconstruct complete 3D assets from partially visible pixels, but it still works only on objects.

## 3   METHOD

Given a set of unposed images $\mathcal{I} = \{\boldsymbol{I}^i\}_{i=1}^K$, ($\boldsymbol{I}^i \in \mathbb{R}^{H \times W \times 3}$) of a scene, our goal is to learn a neural network $\Phi$ that directly produces a complete 3D point cloud, both in terms of *visible* and *occluded* regions. We first discuss the problem formulation in Section 3.1, followed by our 3D latent autoencoder in Section 3.2, and we finally describe our global scene representation in Section 3.3.

### 3.1   PROBLEM FORMULATION

**Problem Definition.**   The input to our model is a set of $K$ *unposed images* $\mathcal{I} = \{\boldsymbol{I}^i\}_{i=1}^K$ of a scene, and the output is a *complete* 3D point cloud $P \in \mathbb{R}^{N \times 3}$, using a feed-forward neural network $\Phi : \mathcal{I} \to P$. This task is conceptually similar to the conventional feed-forward 3D reconstruction setting (Wang et al., 2024a; 2025a; Jiang et al., 2025), except that here $N$ represents the number of points in the *complete* scene point cloud (as shown in Figure 4), rather than $K \times H \times W$ points back-projected from all pixels in each input image.

The *key observation* is that a scene in the real world is composed of a fixed set of physical points, regardless of how many images are captured from different viewpoints. If a physical 3D point is observed in multiple 2D images, the correct representation of the scene should contain a single point, rather than duplicated points back-projected from each observation. Conversely, even if a physical 3D point is never observed in any image, it still exists in the real world and should be inferred by the model. Therefore, the model should be able to predict the occluded regions of the scene and avoid generating redundant points in the overlapping visible regions.

**Data Preprocessing.**   The key to training such a model is the definition of the *complete* 3D point clouds of a scene. It must contain points in both *visible* and *occluded* regions, and avoid duplicated points in the *overlapping visible* regions. The visibility of a 3D point is defined with respect to the input images $\mathcal{I}$. However, the notion of invisible points is ambiguous: there are infinitely many points that are not visible in input images, or even outside the field of view of all input images. To simplify the problem, as shown in Figure 4, we define invisible points within the input-view frustum and discard points outside the frustum.

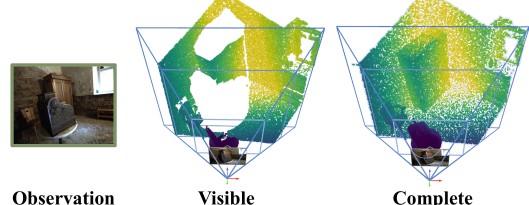

Observation    Visible    Complete

Figure 4: **Visible point clouds *vs.* complete point clouds.** Our NOVA3R aims to recover the complete geometry within the input view's frustum.

Creating such complete point clouds for supervision is non-trivial. The ideal solution is to use the ground-truth 3D mesh of the scene, which can be easily converted to a complete point cloud by uniformly sampling points on the mesh surface. However, the ground-truth mesh is not always available in scene-level datasets. When ground-truth 3D meshes are not available, we instead approximate the complete point clouds using depth maps aggregated from dense views. Specifically, we first back-project the depth maps from all dense views into point clouds, then apply voxel-grid filtering to remove duplicate points in overlapping visible regions. Finally, we discard points outside the frustum of the selected input views (single, two, or a set of views). During training, we apply the farthest point sampling method with random initialization to obtain a subset from the complete point cloud to train our point decoder.

Importantly, as in DUSt3R (Wang et al., 2024a), our complete point clouds are also *view-agnostic*: the 3D points are defined in the coordinate system of the first input view $\boldsymbol{I}_1$, but are *not* pixel-aligned to any input images. This design allows the model to learn to reconstruct the complete 3D scene in

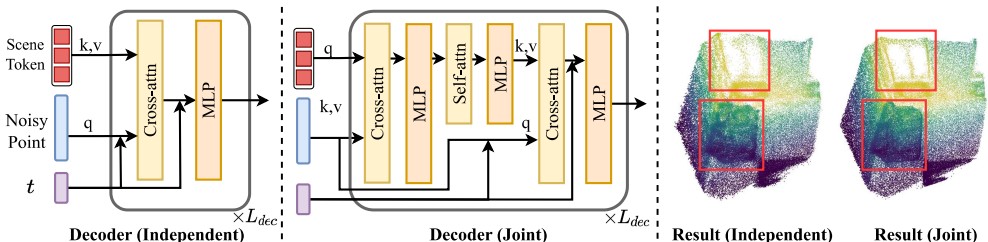

Figure 5: **Different Decoder Architectures.** The independent decoder uses cross-attention only, while the joint decoder implements an efficient self-attention, which yields more precise structures.

the first view's coordinate system while ignoring the ambiguity of pose estimation. Consequently, our model can be trained on a wide range of datasets without requiring ground-truth meshes, unlike existing object-level methods (Zhang et al., 2023; Li et al., 2025b; Team, 2024).

### 3.2 3D LATENT ENCODER-DECODER WITH FLOW MATCHING

Following recent works in 3D latent vector representation (Zhang et al., 2023), we design a 3D latent transformer (Vaswani et al., 2017). However, ours does *not* require a perfect mesh as input or supervision. As shown in Figure 3 (Stage 1), we implement the model as a diffusion model.

**Diffusion-based 3D AutoEncoder.** The encoder $\Phi_{\text{enc}}$ takes the point cloud $P \in \mathbb{R}^{N \times 3}$ as input, and outputs a set of $M$ latent tokens $Z \in \mathbb{R}^{M \times C}$. In practice, to reduce the computational cost, the initial query points $q \in \mathbb{R}^{M \times 3}$ are sampled from the complete point cloud $P \in \mathcal{R}^{N \times 3}$ using farthest point sampling, where $M \ll N$. We further design a hybrid query representation by concatenating the point query with learnable tokens of the same length $M$ along the channel dimension, followed by a linear projection layer that reduces the channel dimension from $2C$ to $C$.

Once the latent tokens $Z$ are obtained, existing 3D VAE methods (Zhang et al., 2023; Team, 2025; Li et al., 2025b) typically use a deterministic decoder to predict an occupancy field $o = \Phi_{\text{dec}}(Z, x)$ or SDF values $s = \Phi_{\text{dec}}(Z, x)$ for each 3D grid query $x \in \mathbb{R}^{N \times 3}$. However, this is not suitable for our task, since obtaining ground-truth occupancy or SDF values for real scene-level datasets is costly or even infeasible. Importantly, unlike objects that can be enclosed within a canonical space, scenes typically lack well-defined boundaries and expand as the number of observations increases, making it difficult to predefine a canonical space. Instead, we directly predict the 3D coordinates of each query point. However, because point clouds are *not* ordered or aligned, we cannot directly map the 3D point query to the ground-truth point clouds $P$ using an $L_2$ loss. We then adopt a diffusion-based decoder $\Phi_{\text{dec}}(x_t, Z, t)$ to decode the scene tokens $Z$ back to the original point cloud space. The transformer-based decoder takes as input a set of $N$ noised query point clouds $x_t \in \mathbb{R}^{N \times 3}$, at the flow matching time $t$, and the latent tokens $Z$ as conditioning. The whole architecture is trained end-to-end with a flow matching loss (Lipman et al., 2023):

$$\mathcal{L}_{\text{flow}}^{\text{AE}} = \mathbb{E}_{t, x_0 \sim P, \epsilon \sim \mathcal{U}(-1,1)} \left[ \| \Phi_{\text{dec}}(x_t, Z, t) - (\epsilon - x_0) \|_2^2 \right], \tag{1}$$

where $x_t = (1 - t)x_0 + t\epsilon$. Note that we do *not* use KL loss or any other regularization on the latent tokens as in existing 3D latent VAE methods (Team, 2025; Li et al., 2025b).

**Architecture.** As noted above, our 3D autoencoder is implemented with a transformer architecture. Specifically, the encoder is built upon TripoSG (Li et al., 2025b), which consists of one cross-attention layer and eight self-attention layers. The decoder has three transformer blocks (details are shown in Figure 5). Notably, the query will be switched between the 3D latent tokens $Z$ and the noisy point clouds $x_t$ in each cross-attention layer. This design reduces the size of the self-attention maps while preserving information flow between latent tokens and query points. Concurrent work (Chang et al., 2024) also proposes a diffusion-based 3D latent autoencoder, but they consider a 3D shape as a probability density function, and process each point independently.

### 3.3 SCENE REPRESENTATION WITH LEARNABLE TOKENS

We now describe how to learn a global scene representation from a set of unposed images. As shown in Figure 3 (Stage 2), we implement it using a large transformer that takes the input images $\mathcal{I}$ and a set of $M$ learnable tokens $t_S \in \mathbb{R}^{M \times C}$ as input, and outputs the scene representation $\hat{Z} \in \mathbb{R}^{M \times C}$.

Table 1: **Quantitative results for scene completion on SCRREAM** (Jung et al., 2024). The *one-side* Chamfer Distance (GT $\rightarrow$ Prediction) results are shown in ( ). $K$ is the number of input views. $*$ denotes methods that are not trained on scene-level data. Our method shows better completion results compared to other competitive baselines. Note that, since NOVA3R is a *non-pixel-aligned* 3D reconstruction model, it does not explicitly distinguish the visible and occluded points.

| Type | Method | Visible ($K$=1) | | | Complete ($K$=1) | | | Complete ($K$=2) | | |
|------|--------|------|------|------|------|------|------|------|------|------|
| | | CD↓ | FS@0.1↑ | FS@0.05↑ | CD↓ | FS@0.1↑ | FS@0.05↑ | CD↓ | FS@0.1↑ | FS@0.05↑ |
| Object | TripoSG* | (0.268) | (0.418) | (0.301) | 0.242 | 0.467 | 0.333 | - | - | - |
| | TRELLIS* | (0.301) | (0.420) | (0.313) | 0.256 | 0.429 | 0.312 | 0.286 | 0.402 | 0.288 |
| Single-view | Metric3D-v2 | 0.063 | 0.803 | 0.534 | 0.086 | 0.725 | 0.473 | - | - | - |
| | DepthPro | 0.055 | 0.852 | 0.603 | 0.079 | 0.764 | 0.535 | - | - | - |
| | MoGe | **0.035** | **0.945** | **0.786** | 0.063 | 0.836 | 0.668 | - | - | - |
| | LaRI | 0.057 | 0.847 | 0.589 | 0.059 | 0.825 | 0.590 | - | - | - |
| Multi-view | DUST3R | 0.059 | 0.851 | 0.653 | 0.086 | 0.757 | 0.565 | 0.061 | 0.833 | 0.641 |
| | CUT3R | 0.069 | 0.835 | 0.679 | 0.091 | 0.753 | 0.543 | 0.092 | 0.739 | 0.532 |
| | VGGT | 0.041 | 0.923 | 0.754 | 0.070 | 0.810 | 0.657 | 0.065 | 0.821 | 0.606 |
| | Ours | (0.043) | (0.904) | (0.730) | **0.048** | **0.882** | **0.687** | **0.053** | **0.862** | **0.657** |

**Learnable Scene Tokens.** As mentioned in Section 3.1, our model aims to predict a *fixed* number of *non-pixel-aligned* points under the first view's coordinate system. Accordingly, in addition to $L$ patchified image tokens $t_I \in \mathbb{R}^{L \times C}$, we introduce a set of $M$ learnable global scene tokens $t_S \in \mathbb{R}^{M \times C}$, which are randomly initialized and optimized during training. The combined token set $t_I \cup t_S$ from all input images, *i.e.*, $t_I = \cup_{i=1}^{K}\{t_I^i\}$, and the learnable scene tokens $t_S$, is fed into a large transformer, with multiple frame- and global-level self-attention layers. To simplify the architecture, the learnable scene tokens $t_S$ are treated as a global frame underlying the first view's coordinate system. This means that the scene tokens undergo the same operations as the image tokens in each Transformer block, except that they use the first view's camera token.

**Architecture.** Our image encoder is built upon VGGT (Wang et al., 2025a), a representative feed-forward 3D reconstruction model. However, we do not use its dense prediction heads to predict the *pixel-aligned* depth and point maps. Instead, we use the output scene tokens $\hat{Z} \in \mathbb{R}^{M \times C}$ as the conditioning of our point decoder $\Phi_{\text{dec}}$, to predict the *non-pixel-aligned* complete 3D point clouds $\hat{P} \in \mathbb{R}^{N \times 3}$. The entire architecture is trained end-to-end with the flow matching loss:

$$\mathcal{L}_{\text{flow}}^{\text{Tran}} = \mathbb{E}_{t,x_0 \sim P, \epsilon \sim \mathcal{U}(-1,1)} \left[ \left\| \Phi_{\text{dec}}(x_t, \hat{Z}, t) - (\epsilon - x_0) \right\|_2^2 \right], \qquad (2)$$

where $\Phi_{\text{dec}}$ is frozen in Stage 2, and only the transformer $\Phi_{\text{tran}} : t_I \cup t_S \rightarrow \hat{Z}$ and the learnable scene tokens $t_S$ are optimized.

# 4 EXPERIMENTS

## 4.1 EXPERIMENTAL SETTINGS

**Metrics.** Following Li et al. (2025a), we report Chamfer Distance (CD) and F-score (FS) at different thresholds (*e.g.,* 0.1, 0.05) for completion tasks. For multi-view reconstruction tasks, we report accuracy (Acc), completion (Comp), and normal consistency (NC) following Wang et al. (2025b). Best results are highlighted as first , second , and third .

**Implementation Details.** By default, we set the number of scene tokens as $M = 768$ and the number of points as $N = 10,000$ for training. The image encoder architecture is exactly the same as VGGT (Wang et al., 2025a), while the 3D latent autoencoder contains 8 layers in the encoder and 3 layers in the decoder. The training contains two stages. In Stage 1, we train the autoencoder for 50 epochs. In Stage 2, we initialize the image encoder with VGGT pretrained weights and the flow-matching decoder with Stage-1 weights, then train for another 50 epochs. Note that, we only fine-tune the image encoder and the scene-token transformer in Stage 2. We train both stages by optimizing the flow-matching loss with the AdamW optimizer and a learning rate of 3e-4. The training runs on 4 NVIDIA A40 GPUs with a total batch size of 32. We use standard flow-matching with cosine noise scheduling, timestep sampling in [0,1], a fixed 0.04 step size at inference, and identical loss settings for both object-level and scene-level datasets.

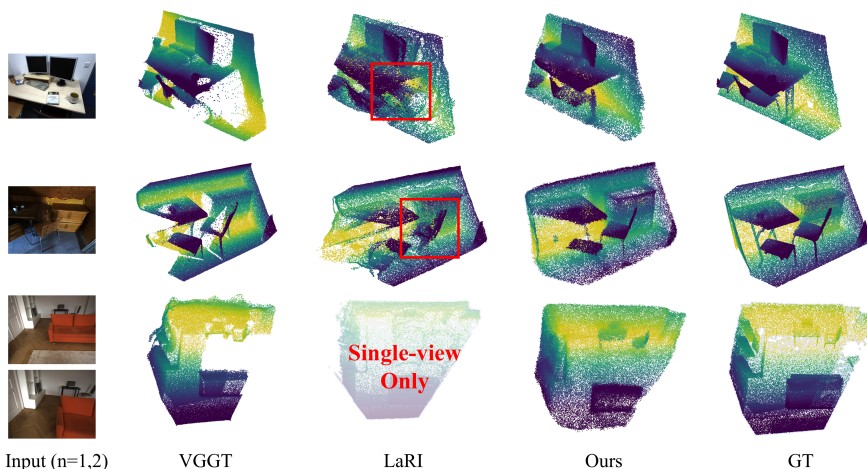

Input (n=1,2)          VGGT          LaRI          Ours          GT

Figure 6: **Qualitative results for scene completion on SCRREAM** (Jung et al., 2024). Our method produces more complete point clouds with clearer and less distorted geometry than other baselines.

Table 2: **Quantitative results for hole area ratio and point cloud density variance on SCR-REAM** (Jung et al., 2024). Our method significantly outperforms pixel-aligned baselines, achieving both lower hole ratios and lower density variance.

| Method | Complete (K=1) | | Complete (K=2) | | Complete (K=4) | |
|---|---|---|---|---|---|---|
| | Hole Ratio↓ | Density Var. ↓ | Hole Ratio ↓ | Density Var. ↓ | Hole Ratio ↓ | Density Var. ↓ |
| DUST3R | 0.317 | 7.758 | 0.237 | 6.553 | 0.257 | 4.801 |
| CUT3R | 0.363 | 8.402 | 0.237 | 6.554 | 0.326 | 4.658 |
| VGGT | 0.307 | 7.105 | 0.238 | 6.546 | 0.261 | 5.217 |
| Ours | **0.088** | **5.127** | **0.121** | **2.188** | **0.134** | **1.881** |

## 4.2 SCENE-LEVEL RECONSTRUCTION

**Datasets.** The scene-level model was trained on 3D-FRONT (Fu et al., 2021) and Scan-Net++V2 (Yeshwanth et al., 2023), using the training splits from LaRI (Li et al., 2025a) and DUSt3R (Wang et al., 2024a), which contain 100k and 230k unique images, respectively. For visible part training, we further incorporate ARKitScenes (Baruch et al., 2021). Ideally, our model is able to handle an arbitrary number of input views, similar to VGGT (Wang et al., 2025a). However, limited by the available computational resources, we mainly verify our contributions on two-view pairs and train with 1–2 input views.

To evaluate the cross-domain generalization ability of our model, we directly evaluate performance on the unseen SCRREAM dataset (Jung et al., 2024), which provides complete ground-truth scans. We follow LaRI's setting for single-view evaluation, with 460 images for testing. For the two-view setting, we sample 329 pairs from the same scene with a frame-ID distance of 40–80, where the maximum pose gap is 30% (measured by point cloud co-visibility) and the hole area ratios (measured by completeness with threshold 0.1) range from 5.3% to 48.6%. We additionally evaluate visible-surface multi-view reconstruction on the 7-Scenes (Shotton et al., 2013) and NRGBD datasets (Azinović et al., 2022), sampling input images at intervals of 100 frames.

**Baselines.** We compare NOVA3R with several representative scene-level 3D reconstruction methods, including **i)** single-view Metric3D-v2 (Hu et al., 2024), DepthPro (Bochkovskiy et al., 2025), and MoGe (Wang et al., 2025c); **ii)** multi-view DUSt3R (Wang et al., 2024a), CUT3R (Wang et al., 2025b), and VGGT (Wang et al., 2025a). However, these methods only focus on *pixel-aligned visible* 3D reconstruction. Hence, we further compare with the concurrent complete 3D reconstruction work LaRI (Li et al., 2025a). Since it does not support multi-view inputs, for completeness, we also report the results from object-level methods TripoSG (Li et al., 2025b) and TRELLIS (Xiang et al., 2025b) by disabling the input mask, while they are not trained on scene-level data.

**Scene Completion.** Following LaRI, we evaluate our amodal 3D reconstruction results on both *visible* and *complete (visible + occluded)* regions. For visible setting, we follow the same evaluation protocol as DUST3R (Wang et al., 2024a) and VGGT (Wang et al., 2025a), where the ground truth contains only the visible points from the input views. For the complete setting, we use the

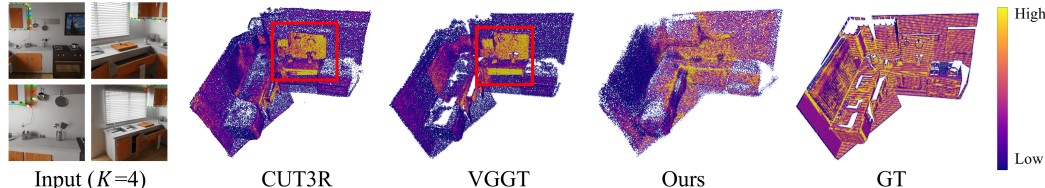

| Input ($K$=4) | CUT3R | VGGT | Ours | GT |

Figure 7: **Qualitative results for density evaluation on NRGBD** ($K = 4$) ([Azinović et al., 2022](#)). Yellow regions denote higher density, and purple regions denote lower density. Despite being trained with only two views, NOVA3R generalizes well to multiple views ($K = 4$).

Table 3: **Quantitative results on visible reconstruction on 7-Scenes** ($K$=2) ([Shotton et al., 2013](#)). Our NOVA3R model can be trained on RGB-D data and achieves competitive results compared to multi-view reconstruction methods. Note that, we use fewer tokens to represent a 3D scene.

| Method | # Tokens | Acc↓ | | Comp↓ | | NC↑ | |
|---|---|---|---|---|---|---|---|
| | | Mean | Med. | Mean | Med. | Mean | Med. |
| DUSt3R | 2048 | 0.054 | 0.023 | 0.075 | 0.034 | 0.772 | 0.901 |
| Spann3R | 784 | 0.044 | 0.022 | 0.046 | 0.025 | 0.792 | 0.922 |
| CUT3R | 768 | 0.043 | 0.023 | 0.054 | 0.028 | 0.760 | 0.884 |
| VGGT | 2738 | 0.042 | **0.020** | 0.045 | 0.025 | **0.813** | **0.923** |
| Ours | 768 | **0.041** | 0.021 | **0.033** | **0.019** | 0.794 | 0.917 |

full point cloud as ground truth, including occluded and unseen regions. However, unlike pixel ray-conditional LaRI, NOVA3R does not explicitly identify the visible region. We therefore adopt *one-sided* Chamfer Distance (GT → Prediction) for the visible part: each GT-visible point must be explained by a nearby prediction. This measures coverage of the visible ground truth, yet without penalizing missing, occluded regions. Table 1 shows three settings: 1-view *visible*, 1-view *complete*, and 2-view *complete*. Despite using only two datasets to train, our method consistently outperforms multi-view baselines on complete reconstruction in both $K = 1$ and $K = 2$ settings, demonstrating the effectiveness of our non–pixel-aligned approach. Our method also achieves competitive results on visible-surface reconstruction. Qualitative results in Figure 6 show that our method produces surfaces without holes (unlike pixel-aligned methods such as VGGT) and yields clearer, less distorted geometry than LaRI. This benefit is attributed to our non–pixel-aligned design, which prevents ray-direction bias in reconstruction. We further quantify the completion capability using the hole area ratio, which is computed by checking whether each ground-truth point has a predicted point within a distance threshold of 0.1. As shown in Table 2, our method consistently achieves significantly lower hole ratios, demonstrating its strong capability for complete reconstruction. In terms of density variance, our approach outperforms all pixel-aligned baselines, even in unseen four-view settings, indicating better physical plausibility with more evenly distributed point clouds. Moreover, when comparing across different $K$, the density variance consistently decreases from one to four input views, further confirming that incorporating more views leads to improved spatial uniformity.

**Physically-plausible Reconstruction.** Beyond 3D completion, our *non–pixel-aligned* formulation also features physically plausible reconstruction by fusing evidence in 3D rather than along camera-pixel rays, reducing duplicated points in overlapping regions and improving cross-view consistency. To illustrate this, we evaluate visible reconstruction with $K = 4$ views on NRGBD ([Azinović et al., 2022](#)). As shown in Figure 7, pixel-aligned methods like CUT3R ([Wang et al., 2025b](#)) and VGGT ([Wang et al., 2025a](#)) accumulate 3D points in co-visible regions, producing uneven densities and multi-layer artifacts. This is physically incorrect, as each point corresponds to a single location in the real world, regardless of the number of views. In contrast, our NOVA3R generates cleaner, single-surface geometry with uniform point distribution, achieving competitive results despite using fewer datasets and views (see Table 3). We further quantify physical plausibility by computing the density variance in Table 2, which indicates that our method achieves a more uniformly distributed reconstruction compared to pixel-aligned baselines.

### 4.3 OBJECT-LEVEL RECONSTRUCTION

**Datasets.** We demonstrate the versatility of our method as a unified non–pixel-aligned approach for both scenes and objects. Following [Li et al. (2025a)](#), we train an object-completion model on Objaverse ([Deitke et al., 2023](#)) with 190k annotated images. For evaluation, we report results on

Table 4: **Quantitative results for object completion on GSO (Downs et al., 2022).** NOVA3R provides a unified solution for both scene and object completion from unposed images.

| Type | Method | View-aligned ($K$=1) | | | View-aligned ($K$=2) | | |
|------|--------|--------|--------|--------|--------|--------|--------|
| | | CD ↓ | FS@0.1 ↑ | FS@0.05 ↑ | CD ↓ | FS@0.1 ↑ | FS@0.05 ↑ |
| Single-view | SF3D | 0.037 | 0.913 | 0.738 | - | - | - |
| | SPAR3D | 0.038 | 0.912 | 0.745 | - | - | - |
| | LaRI | 0.025 | 0.966 | 0.894 | - | - | - |
| | TripoSG | 0.025 | 0.961 | 0.899 | - | - | - |
| Multi-view | TRELLIS | 0.025 | 0.962 | 0.896 | 0.028 | 0.946 | 0.874 |
| | Ours | **0.020** | **0.985** | **0.925** | **0.023** | **0.978** | **0.903** |

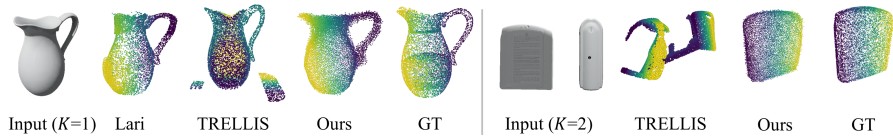

| Input ($K$=1) | Lari | TRELLIS | Ours | GT | Input ($K$=2) | TRELLIS | Ours | GT |

Figure 8: **Qualitative results for object completion on GSO (Downs et al., 2022).** Our method provides more precise geometry and better 3D consistency with multi-view inputs.

unseen Google Scanned Objects (Downs et al., 2022). For single-view reconstruction, we use the same 1030-object split as LaRI (Li et al., 2025a). For two-view reconstruction, we fix the 0th image and uniformly sample three additional views, yielding three pairs per object (3090 pairs in total).

**Baselines.** We compare with several representative object-level 3D reconstruction methods, including SF3D (Boss et al., 2025), SPAR3D (Huang et al., 2025), TripoSG (Li et al., 2025b), and TRELLIS (Xiang et al., 2025b). We also include LaRI (Li et al., 2025a) as a strong baseline, which is trained on the same dataset and supports amodal 3D reconstruction.

**Object Completion.** Table 4 reports results for single view ($K = 1$) and two views ($K = 2$). Our NOVA3R outperforms LaRI on all three metrics. Importantly, our pipeline supports multi-view completion that maps different unposed images into the same view-aligned space. On the multi-view benchmark, our method also outperforms TRELLIS, highlighting the benefits of non–pixel-aligned reconstruction for consistent global geometry. Qualitative comparisons in Figure 8 show that our completions preserve fine structures, and achieve better 3D consistency in the multi-view setting.

## 4.4 ABLATION STUDIES

We perform comprehensive ablation studies on the SCRREAM complete ($K = 1$) setting to validate the key design choices of our method, with particular emphasis on assessing the contribution of Scene Tokens to global structure modeling. The results are summarized in Table 5, and we discuss each component in detail below.

**Initial Query (Stage 1).** Prior work (Zhang et al., 2023) shows that the initialization of point queries affects autoencoder performance. We compare three options: (i) *downsampled input points*, (ii) *learnable query tokens*, and (iii) a *hybrid* that concatenates (i) and (ii). Downsampled points preserve the input geometry distribution, whereas learnable tokens add flexibility under source–target shifts. As shown in Table 5, the hybrid combines these benefits and yields the best results.

**Number of latent scene tokens (Stage 1).** As described in Section 3, we represent each scene with a fixed-length set of latent tokens. The number of tokens $M$ directly affects the representation capacity and ability to capture fine details, especially in large scenes. We evaluate different numbers of scene tokens from $\{256, 512, 768\}$ and observe consistent improvements as the count increases (see Table 5). To balance accuracy and efficiency, we use $M = 768$ tokens by default. Ideally, $M$ could be further increased for better performance. We leave this for other works to explore.

**Different architecture of flow-matching decoder (Stage 1).** The latest work (Chang et al., 2024) also presents a flow-matching decoder for point cloud encoder, but it assumes that all points are independent (shown in Figure 5). This design is efficient, but ignores spatial correlations between points. In our work, we instead adopt a lightweight *self-attention + cross-attention* decoder that jointly reasons over points and scene tokens, allowing information exchange across the point set.

Table 5: **Ablations.** All models are evaluated on the SCRREAM complete ($K = 1$) setting. We report CD↓, FS@0.1↑, FS@0.05↑ and FS@0.02↑ across different ablation settings.

| Settings | Initial tokens (Stage 1) | | | # Scene tokens (Stage 1) | | | FM Decoder (Stage 1) | | Img Resolution (Stage 2) | |
|---|---|---|---|---|---|---|---|---|---|---|
| | Point | Learnable | **Hybrid** | 256 | 512 | **768** | Indep. | **Joint** | 224 | **518** |
| CD↓ | 0.011 | 0.013 | **0.011** | 0.014 | 0.013 | **0.011** | 0.012 | **0.011** | 0.054 | **0.048** |
| FS@0.1↑ | 0.999 | 0.998 | **0.999** | 0.996 | 0.998 | **0.999** | 0.998 | **0.999** | 0.861 | **0.882** |
| FS@0.05↑ | 0.991 | 0.981 | **0.993** | 0.975 | 0.986 | **0.993** | 0.990 | **0.993** | 0.648 | **0.687** |
| FS@0.02↑ | 0.894 | 0.841 | **0.904** | 0.811 | 0.839 | **0.904** | 0.889 | **0.904** | 0.327 | **0.350** |

Table 6: **Ablations on different training loss functions.** All models are evaluated on the SCRREAM complete ($K = 1$) setting. We report CD↓, FS@0.1↑, FS@0.05↑ and FS@0.02↑ and inference time↓ for the decoder.

| Training Loss | SCRREAM (Stage 1) | | | | |
|---|---|---|---|---|---|
| | CD ↓ | FS@0.1 ↑ | FS@0.05 ↑ | FS@0.02 ↑ | Inference Time (s) ↓ |
| Chamfer distance | 0.024 | 0.981 | 0.907 | 0.575 | **0.557** |
| Flow-matching | **0.011** | **0.999** | **0.993** | **0.904** | 2.985 |

To investigate the effect of this design, we compare it with an independent variant without self-attention. Empirically, the joint decoder yields lower CD, higher F-scores, and sharper fine details (Table 5), with small quantitative but significant qualitative gains (Figure 5).

**Input image resolution (Stage 2).** In Stage 2 (image-to-point), we adopt a transformer to integrate information between image and scene tokens. The input resolution determines the number of image tokens used in the aggregation process. With patch size 14, a resolution of $224 \times 224$ yields $16 \times 16 = 256$ tokens, while a resolution of $518 \times 518$ yields $37 \times 37 = 1369$ tokens. As shown in Table 5, training with $518 \times 518$ resolution consistently improves CD and F-scores.

**Flow-matching loss vs. Chamfer distance loss.** To verify the necessity of flow-matching for unordered point cloud encoding, we conduct an ablation using the same architecture but replacing the flow-matching loss with Chamfer Distance. Both models were trained on SCRREAM (Stage 1) under the same protocol. As shown in Table 6, flow-matching achieves significantly better reconstruction quality and generalization. Chamfer Distance struggles in scene-level settings because its nearest-neighbor formulation is computationally expensive, sensitive to density imbalance, and unable to capture global structure across varying scales and input views, while flow-matching produces stable, complete, and globally consistent reconstructions.

## 5 CONCLUSION

We present NOVA3R, a non-pixel-aligned framework for amodal 3D scene reconstruction from unposed images. Unlike prior pixel-aligned methods, our NOVA3R achieves state-of-the-art results in amodal 3D reconstruction, including both visible and invisible points, on both scene and object levels. Notably, it pioneers a new paradigm for physically plausible scene reconstruction that reconstructs a uniform point cloud for the entire scene, without holes or duplicated points. This simple yet effective design makes it a promising solution for real-world applications.

**Limitations and Discussion.** Due to computational constraints, we train our model with a relatively small number of scene tokens and point clouds and a moderate number of input views (up to 2). Hence, the reconstruction quality may degrade for large-scale scenes with complex structures. Future work can explore scaling up the model and training data to enhance performance and generalization. In addition, our model currently focuses on reconstructing static scenes and does not handle dynamic objects or temporal consistency across frames.

### ACKNOWLEDGMENTS

We would like to thank Ruining Li, Zeren Jiang, and Brandon Smart for their insightful feedback on the draft. This work was supported by the ERC Advanced Grant "SIMULACRON" (agreement #884679), the GNI Project "AI4Twinning", and the DFG project CR 250/26-1 "4DYoutube". Chuanxia Zheng is supported by NTU SUG-NAP and National Research Foundation, Singapore, under its NRF Fellowship Award NRF-NRFF172025-0009.

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

## A APPENDIX

### A.1 MORE IMPLEMENTATION DETAILS.

**Model architectures.** For the 3D point autoencoder (Stage 1), we follow the point encoder design from TripoSG Li et al. (2025b), which consists of one cross-attention layer and eight self-attention layers. The initial point queries are obtained by farthest point sampling from the input point cloud, while the learnable queries are randomly initialized tokens. We use 768 tokens with dimension 128 for the our model. For the flow-matching decoder, we use a joint block with two cross-attention layers and one self-attention layer. The goal is to enable self-attention–like information exchange among queries while keeping computation manageable. Concretely, each block (i) aggregates information from noisy query points into the scene tokens via cross-attention, (ii) performs self-attention on the scene tokens (small $M$) to mix global context efficiently, and (iii) projects the updated scene tokens back to the queries with a second cross-attention.

For the image-to-latent transformer in Stage 2, we follow the architecture of VGGT (Wang et al., 2025a), which alternates between local (frame-level) and global attention. Due to computational constraints, we adopt a 16-layer variant instead of the full 24-layer VGGT, initializing from its pretrained weights. We also reuse VGGT's image tokenizers with frozen weights to obtain image tokens. The initial 3D scene tokens are treated as a *3D frame* and share the same local attention mechanism with the image tokens. For the 3D scene token, we copy the camera token from the first view to enable reconstruction in the camera coordinate of the first view.

**Training.** We train our model in two stages. In Stage 1, we aggregate per-view point clouds into a single input cloud and apply farthest point sampling on a randomly selected subset to supervise the flow-matching decoder. Farthest point sampling ensures that the target point cloud is distributed more evenly, reducing the influence of overlapping points in visible regions. Stage 1 is trained for 50 epochs. In Stage 2, we reuse the flow-matching decoder from Stage 1 and train it together with our image encoder, initialized with pretrained VGGT weights. The same flow-matching loss is used in both stages. For object and scene completion, target point clouds are sampled from complete reconstructions. To demonstrate compatibility with pixel-aligned formats, we also train a variant using RGB-D input, where target point clouds are sampled from point maps back-projected from depth. Stage 2 is trained for another 50 epochs.

Regarding computational cost, the Stage-1 point encoder is lightweight and requires no paired image–point cloud data, enabling efficient training on large-scale synthetic 3D datasets. In practice, Stage 1 takes about 40% less training time than Stage 2, and inference remains single-stage, feedforward, and efficient regardless of the two-stage setup. Overall, the two-stage design adds small overhead while substantially improving stability, data flexibility, and reconstruction quality.

**Evaluation.** For object- and scene-level completion, we follow Li et al. (2025a) and sample 10k points for the object task and 100k for the scene task. However, correspondence-based point cloud alignment is not applicable due to our non–pixel-aligned reconstruction. Instead, we optimize a 3D translation and a global (1D) scale relative to the ground-truth point cloud using Adam to improve alignment. We do not optimize rotation, as our reconstruction is expressed in the first-view coordinate frame.

### A.2 MORE ABLATION STUDY.

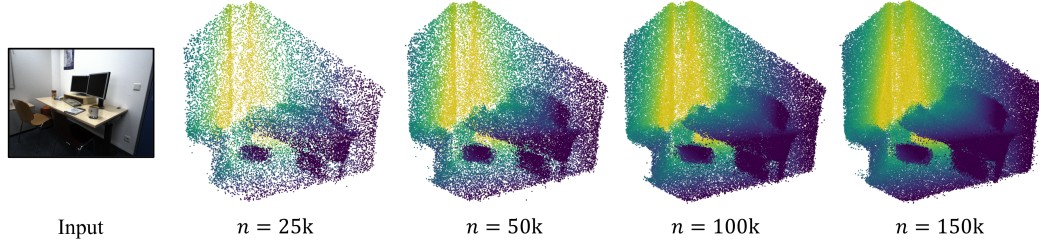

Figure 9: **Visualization of point cloud generation at different resolutions.** Our non–pixel-aligned formulation allows inference at arbitrary resolutions.

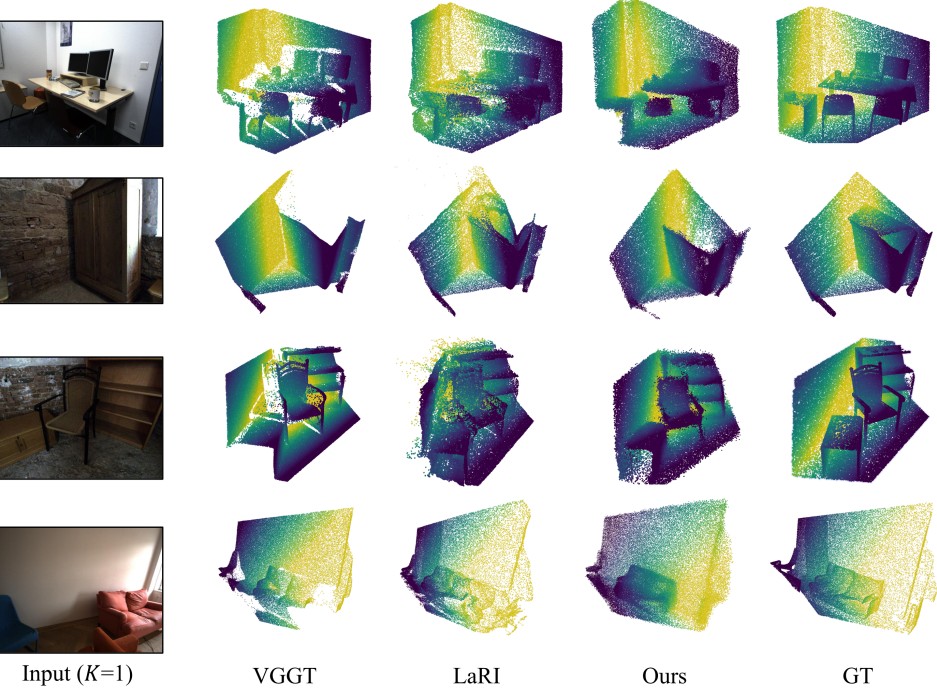

Input (*K*=1)  VGGT  LaRI  Ours  GT

Figure 10: **Qualitative results for scene completion on SCRREAM Jung et al. (2024).** Our method shows better scene completion results compared to other baselines.

**Reconstruction at any resolution.** Since NOVA3R models a point distribution rather than a per-pixel point map, it naturally supports resolution-agnostic generation by adjusting the number of noisy queries at inference. Figure 9 presents results with varying query counts for the flow-matching decoder, demonstrating that our method consistently produces point clouds at different resolutions with reliable reconstruction quality.

### A.3 MORE VISUALIZATIONS.

We show more visualization results for scene-level completion on SCRREAM (Jung et al., 2024) dataset, as shonw in Figure 10. We also include density evaluation on NRGBD (Azinović et al., 2022) in Figure 11. While trained with $K = 2$ views only, our method generalize to multiple image views ($K = 4$) and provides more evenly distributed point cloud .

### A.4 REDUCING UNCERTAINTY IN LATENT DIFFUSION–BASED 3D GENERATION

Our method is specifically designed to reduce the uncertainty typically observed in latent diffusion–based 3D generation approaches such as TRELLIS (Xiang et al., 2025a) and TripoSG (Li et al., 2025b). These methods perform generation in a high-dimensional latent space, which may lead to hallucinated geometry, shape deviations, and inconsistencies across viewpoints—particularly when multiple input images are involved. As a result, they struggle to maintain strong pixel-to-scene and cross-view alignment.

In comparison, NOVA3R provides faithful reconstruction conditioned on the input images. Furthermore, NOVA3R can be integrated with the pretrained TRELLIS model to provide active voxel positions, effectively extending 3D object generation models to real-world scene synthesis without re-training (see Figure 12).

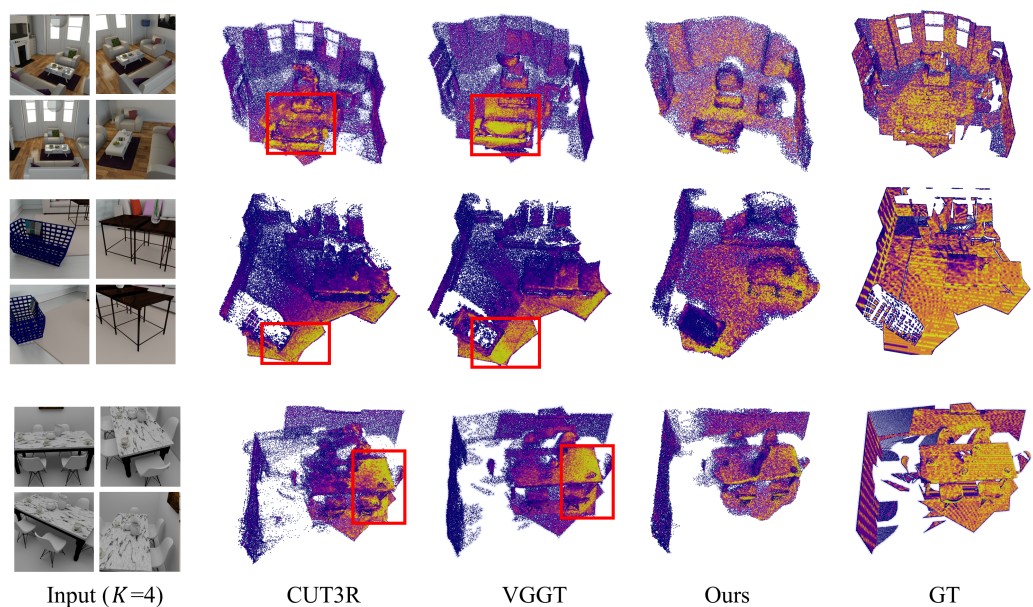

| Input ($K$=4) | CUT3R | VGGT | Ours | GT |

Figure 11: **Qualitative results for density evaluation on NRGBD** ($K$=4) (**Azinović et al., 2022**). Yellow regions denote higher density, and purple regions denote lower density. Our method provides more evenly-distributed point cloud (colored by density).

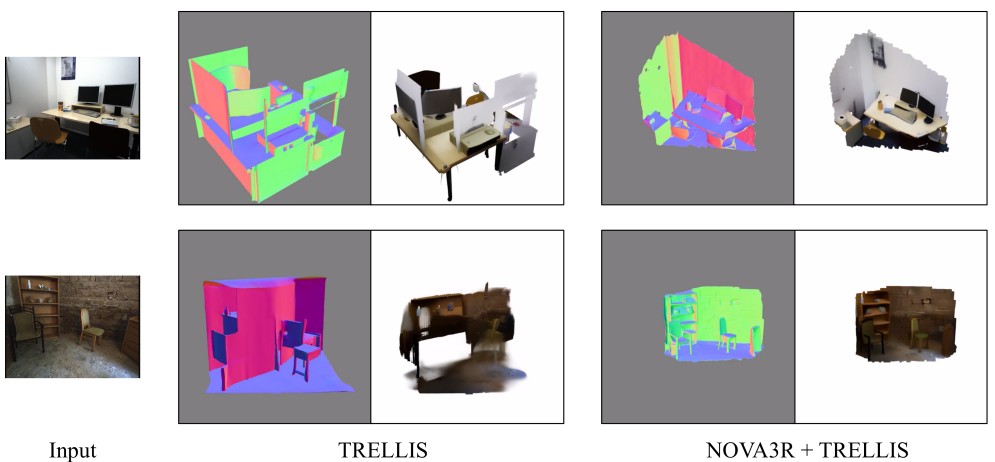

| Input | TRELLIS | NOVA3R + TRELLIS |

Figure 12: **Extending 3D object generation model to real-world scene reconstruction.** The pre-trained TRELLIS model struggles to generate geometrically faithful reconstructions for real-world scenes. In contrast, NOVA3R can provide active voxel priors for the TRELLIS stage-1 generation process, enabling its extension to real-world scene synthesis.

### A.5 Performance on Outdoor Scenes

To validate the robustness and generalization capability of our framework, we further evaluate NOVA3R using the outdoor dataset Virtual KITTI 2 (Cabon et al., 2020). We finetune our model on Virtual KITTI 2 to better adapt to large-scale outdoor environments. To construct pseudo ground

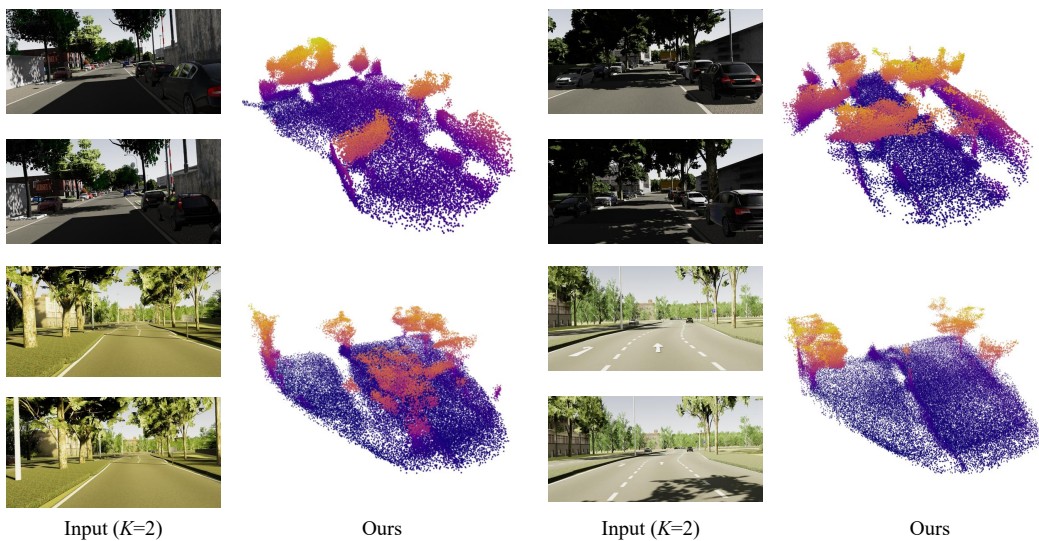

Input (*K*=2)        Ours        Input (*K*=2)        Ours

Figure 13: **Qualitative results for outdoor scenes reconstruction on Virtual KITTI 2.** Our method is also applicable to outdoor scene reconstruction (colored by Y axis).

truth, for each input frame we collect neighboring frames within [-4,8] timesteps and additional views from $\pm 15°$ and $\pm 30°$ viewpoints. Using depth maps and camera parameters, we project them into per-frame point clouds, transform them to world coordinates, and retain only points within the target view's frustum. As shown in Figure 13, NOVA3R performs well on outdoor scenes, further demonstrating its ability to handle both indoor and outdoor scenarios.

## A.6 DISCUSSION

**Large-scale Scenes.** Modeling large-scale scenes with many input images is a major computational bottleneck for existing learning-based 3D reconstruction methods, particularly for pixel-aligned approaches like VGGT, which must handle duplicated points across multiple views. In contrast, our point-wise decoding uses fewer tokens to represent the scene, making it inherently more scalable. However, the number of points needed varies across scenes of different scales, requiring adaptive point selection strategies, such as using sparse COLMAP point clouds as guidance.

**Dynamic Scenes.** Our paradigm is inherently extensible to dynamic scenes, either by adding a branch to predict target time point maps (Sucar et al., 2025; Feng et al., 2025) or by extending the 3D latent autoencoder to a time-conditioned 4D latent representation. Such a representation can potentially model the entire 4D scene more efficiently by capturing both complete geometry and temporal evolution across the whole sequence, rather than relying on per-frame reconstruction.

