# OpenReview forum: "NOVA3R: Non-pixel-aligned Visual Transformer for Amodal 3D Reconstruction"
_ICLR.cc/2026/Conference — ICLR 2026 Poster_

### Official Review · Reviewer_t2bA · 2025-10-27

**Soundness:** 4
**Presentation:** 3
**Contribution:** 4
**Rating:** 6
**Confidence:** 4

**Summary:**

This paper introduces NOVA3R, a novel feed-forward framework for amodal 3D reconstruction from a set of unposed images. The central problem the authors address is the limitations of traditional "pixel-aligned" methods (like DUSt3R or VGGT), which are tied to per-ray predictions. NOVA3R's core idea is to decouple reconstruction from pixel alignment by learning a global, view-agnostic scene representation. This allows the model to recover both visible and occluded points and generate a single, physically plausible point cloud for the entire scene. The method is implemented as a two-stage pipeline, the first is 3D Latent AutoEncoder training and the second one is Feed-forward Scene Prediction. Experiments on both scene-level (SCRREAM, NRGBD) and object-level (GSO) datasets show that NOVA3R outperforms state-of-the-art methods in reconstruction completeness and accuracy.

**Strengths:**

1. The paper's main strength is its "non-pixel-aligned" formulation. This is a clear and well-motivated departure from dominant pixel-aligned methods. It directly addresses fundamental, known issues in 3D reconstruction (incompleteness and duplicate geometry).
2. The architectural design is novel and effective. Using a diffusion decoder with a flow-matching loss elegantly transforms noisy point clouds to target distribution with conditioning.
3. The experimental validation is thorough. NOVA3R demonstrates superior performance over strong baselines in its primary goal of complete amodal reconstruction (Table 1) and also shows it can be applied to object-level tasks (Table 3).
4. The paper shows that this new paradigm is also efficient. NOVA3R has significantly fewer parameters (718M) than both the pixel-aligned VGGT (941M) and the latent-generation TRELLIS (1785M) models.

**Weaknesses:**

1. While the method commendably avoids the need for high-quality meshes, the autoencoder in Stage 1 still requires "complete point clouds" for supervision. The authors' solution is to aggregate depth maps from dense views. However, it still can not eliminate the ambiguity of occluded or invisible area.
2. The point cloud is noisier compared to VGGT. VGGT presents a smoother plane in Figure 6. This is also reflected in normal consistency metrics.
3. Static scene only. It seems like this paradigm is hard to generalize to dynamic scenes.

**Questions:**

1. In stage1's diffusion decoder training, how do you matches the unordered points in flow matching?
2. Any experiments on outdoor scenes?

---

> ### Author Response · Authors · 2025-11-20
> **Official Response by Authors (Part 1)**
>
> We sincerely thank the reviewer for the insightful comment and constructive feedback. We have carefully addressed each comment and hope that our responses clarify the concerns. We would be glad to further elaborate if any additional questions remain.
>
> > **[W1] The autoencoder in Stage 1 still requires "complete point clouds" for supervision. The authors' solution is to aggregate depth maps from dense views. However, it still can not eliminate the ambiguity of occluded or invisible area.**
>
> Unlike *object-level* 3D datasets where watertight meshes are often available (like Objaverse-XL dataset), *scene-level* data typically lacks perfect meshes,
> making it challenging to directly training 3DShape2VecSet-style autoencoders.
> However, with sufficient multi-view coverage, aggregating depth maps from dense viewpoints can yield a highly complete approximation of the underlying geometry, as demonstrated in datasets like ScanNet.
> Our approach leverages this property to generate pseudo-complete point clouds,
> for training the Stage-1 autoencoder, rather than relying on perfect meshes.
>
> More importantly, a key advantage of our design is its ability to flexibly train on point clouds with varying ground truth completeness, ranging from partial to complete, without the need for watertight meshes.
> This makes our approach more adaptable to real-world scenarios where perfect completeness is rarely attainable for *scene-level* data.
>
> > **[W2] The point cloud is noisier compared to VGGT. VGGT presents a smoother plane in Figure 6. This is also reflected in normal consistency metrics.**
>
> We believe the relatively noisier appearance is mainly due to the significant difference of data scale and diversity between training sets, rather than a fundamental limitation of our approach.
> In particular, VGGT is trained on **17 large-scale RGB-D** and multi-view datasets, covering a wide range of scene types, geometric structures, and depth-supervised signals, which naturally promotes smoother surfaces and stronger normal consistency.
>
> In contrast, our model is trained on considerably fewer datasets **3DFront and Scannet++** and without large-scale RGB-D supervision, yet still achieves competitive normal consistency and strong geometric fidelity (see Table 1, main paper).
> In addition, NOVA3R uses significantly fewer parameters (718M) compared to the pixel-aligned VGGT (941M).
> We expect that scaling our training to similarly diverse RGB-D datasets would further reduce surface noise, making this an empirical limitation rather than a methodological one.
>
> > **[W3] Static scene only. It seems like this paradigm is hard to generalize to dynamic scenes.**
>
> This paradigm is originally designed to be applicable to dynamic scenes by either
> - adding additional branches to predict the target time point maps (similar to Dynamic Point Maps, ST4RTack, and Stereo4D)
> - extending the 3D latent autoencoder to a 4D latent representation conditioned on time.
>
> Actually,
> this representation may be more efficient for dynamic scenes,
> as it can model the 4D scene holistically--a complete 3D geometry and track its motion--from the beginning to the end of the sequence,
> rather than per-frame reconstruction.
> However,
> we need to carefully design the training data and loss functions to handle temporal coherence and motion ambiguity,
> which is beyond the current scope of this work.
> Here,
> we primarily focus on scene-level completion for static scenes from multi-view inputs, which remains a challenging and under-explored problem.
> Another discussion of this extension is also provided in our reply to **[PSme-Q4]**.

---

> ### Author Response · Authors · 2025-11-20
> **Official Response by Authors (Part 2)**
>
> > **[Q1] In stage1's diffusion decoder training, how do you matches the unordered points in flow matching?**
>
> A key benefit of using flow matching is that it does not require any predefined correspondences between unordered point sets.
> Instead, correspondences are implicitly established during the flow-matching process, which learns to map from the noisy distribution to the target point cloud distribution.
> Specifically, we generate a clean target point cloud $x_0$ via random downsampling of the ground truth point cloud, and construct the noised point cloud $x_t$ by interpolation between $x_0$ and random noise $\epsilon$ using $x_t = (1-t)x_0 + t \epsilon$.
> Because $x_t$ and $x_0$ share point indices, the $(x_0, x_t)$ pair naturally provides correspondences for supervision.
> This enables the model to learn a continuous mapping from the noise distribution to the target point cloud distribution conditioned on the scene tokens, without requiring explicit point-level correspondence.
>
> > **[Q2] Any experiments on outdoor scenes?**
>
> To validate the robustness and generalization capability of our framework, we further evaluate NOVA3R using the outdoor dataset Virtual KITTI 2.
> We finetune our model on Virtual KITTI 2 to better adapt to large-scale outdoor environments.
> To construct pseudo ground truth, for each input frame we collect neighboring frames within [-4,8] timesteps and additional views from $\pm 15^{\circ}$ and $\pm 30^{\circ}$ viewpoints.
> Using depth maps and camera parameters, we project them into per-frame point clouds, transform them to world coordinates, and retain only points within the target view’s frustum.
> As illustrated in Figure 13 (Supplementary), NOVA3R produces reasonable outdoor reconstructions, demonstrating its applicability to both indoor and outdoor scenarios.

---

### Official Review · Reviewer_Fedf · 2025-10-30

**Soundness:** 3
**Presentation:** 3
**Contribution:** 3
**Rating:** 6
**Confidence:** 4

**Summary:**

NOVA3R is a non-pixel-aligned visual Transformer approach for feed-forward amodal 3D reconstruction from unposed images. It addresses the limitations of pixel-aligned methods—such as only reconstructing visible regions and generating duplicated structures in overlapping areas—and overcomes the inability of latent 3D generation methods to handle complex scenes by learning a global, view-agnostic scene representation.

The core of this method includes a scene-token mechanism (to aggregate multi-view information) and a diffusion-based 3D decoder (to generate complete point clouds). It adopts a two-stage training process: first, a 3D latent autoencoder is trained using a flow-matching loss; then, a pre-trained image encoder and learnable scene tokens are used to map images to the latent space.

Experiments show that NOVA3R outperforms state-of-the-art methods on both scene-level (SCRREAM, 7-Scenes) and object-level (GSO) datasets, producing more complete and physically plausible reconstruction results. It supports both single-view and multi-view inputs, combining feed-forward efficiency with strong modeling capabilities. Its limitations include insufficient reconstruction quality for large-scale complex scenes and no support for dynamic objects.

**Strengths:**

1. By modeling the entire 3D reconstruction task as a two-stage generative process, NOVA3R breaks the traditional pixel-aligned paradigm that estimates geometric attributes by tying geometry to per-ray predictions. This innovative design decouples reconstruction from pixel alignment, enabling the model to learn a global, view-agnostic scene representation and thus reconstruct point clouds of both visible and invisible (occluded) regions, addressing the incompleteness limitation of pixel-aligned methods .

2. NOVA3R innovatively adopts diffusion as the point cloud decoder, which is highly suitable for the inherent characteristics of point clouds (e.g., unordered and unaligned nature). Unlike deterministic decoders that struggle with matching unordered point sets, the diffusion-based decoder, trained with a flow-matching loss, effectively resolves matching ambiguities in unordered point clouds, ensuring the generation of complete, physically plausible non-pixel-aligned point clouds without duplicated structures.

3.experiments

**Weaknesses:**

1. Compared with end-to-end architectures, NOVA3R requires additional training efforts due to its two-stage design. The model first trains a 3D latent autoencoder (Stage 1) with a flow-matching loss to compress complete point clouds into latent tokens and decode them, then optimizes the image encoder and learnable scene tokens (Stage 2) to map unposed images to the latent space. This two-stage training not only increases the overall training pipeline complexity but also demands more computational resources and time compared to one-step end-to-end reconstruction methods.

2. Concerns arise regarding the consistency between the reconstructed point clouds and input images under NOVA3R’s paradigm. Unlike pixel-aligned methods that directly predict geometry attributes (e.g., depth, point maps) tied to image pixels, NOVA3R undergoes a "compression-decompression" process: it compresses scene information into latent tokens and decompresses them via a diffusion-based generative decoder. This indirect mapping, combined with the stochastic nature of diffusion, introduces uncertainties about whether the generated non-pixel-aligned point clouds can fully align with the visual content of input images, requiring further validation of cross-view and pixel-scene consistency.

3. This modeling approach may struggle to scale to scenarios with an extremely large number of input images, primarily due to computational bottlenecks caused by its point-wise decoding nature. As the number of point clouds required for reconstructing complex, large-scale scenes increases (e.g., when processing more input views to enhance reconstruction completeness), the diffusion-based decoder— which operates on individual points during the decompression process—faces a significant surge in computational load. Unlike pixel-aligned methods that can leverage per-ray parallelization tied to image pixels, NOVA3R’s non-pixel-aligned, point-wise decoding lacks efficient scalability for handling massive point volumes, making it challenging to adapt to tasks requiring extensive multi-view input or high-density point cloud output.

**Questions:**

Regarding the paper proposing NOVA3R, there are three key questions to raise with the authors, corresponding to potential weaknesses: First, compared with end-to-end structures, the two-stage training design requires additional training efforts, and it is worth discussing whether the training complexity can be optimized. Second, the paradigm of "compression-decompression" plus diffusion-based decoding may bring uncertainty to the consistency between the reconstructed point cloud and the input image, and further verification of this consistency is needed. Third, the point-wise decoding nature makes it difficult for the model to scale to scenarios with an extremely large number of input images, as the increase in point cloud quantity will lead to significant computational bottlenecks.

Nevertheless, integrating the generative idea into the VGGT series of works is an excellent and valuable concept. Although a hybrid framework (using VGGT for visible regions and the proposed method for invisible regions) might perform better than solely adopting NOVA3R, the framework proposed in this paper still has significant reference value. Therefore, it is recommended to give this paper a borderline accept decision.

---

> ### Author Response · Authors · 2025-11-20
> **Official Response by Authors (Part 1)**
>
> We sincerely thank the reviewer for the insightful comment and constructive feedback. We have carefully addressed each comment and hope that our responses clarify the concerns. We would be glad to further elaborate if any additional questions remain.
>
> > **[W1] This two-stage training not only increases the overall training pipeline complexity but also demands more computational resources and time compared to one-step end-to-end reconstruction methods.**
>
>
> While end-to-end regression-based training is more straightforward, it often fails to handle invisible regions, leading to geometrically distorted results (e.g., in VGGT and CUT3R). To address this, two-stage training has been widely adopted in latent 3D generation, but mostly at the object level and requiring perfect meshes. In contrast, our two-stage design is the first to apply this paradigm to scene-level reconstruction and completion, allowing training on large-scale synthetic 3D data without paired image–point cloud supervision. By learning the flow-matching decoder independently, Stage 2 training becomes much more stable and converges faster, since the image encoder only needs to map images into a well-structured latent space.
>
> Regarding computational cost, the Stage-1 point encoder is lightweight and requires no paired image–point cloud data, allowing efficient training on large-scale synthetic 3D datasets. In practice, Stage 1 takes approximately 40\% less training time than Stage 2. Moreover, inference remains single-stage, feed-forward, and efficient, regardless of the two-stage training scheme. Therefore, the two-stage approach introduces minimal additional cost while significantly improving stability, data flexibility, and reconstruction quality. In total, our method was trained on 4 NVIDIA A40 GPUs for 4 days (Stage 1 + Stage 2), which requires significantly lower computational cost compared to VGGT (trained on 64 A100 GPUs for 9 days) and TripoSG (trained on 160 A100 GPUs for 3 weeks).
>
> > **[W2]  This indirect mapping, combined with the stochastic nature of diffusion, introduces uncertainties about whether the generated non-pixel-aligned point clouds can fully align with the visual content of input images.**
>
> Our method is specifically designed to reduce the uncertainty typically observed in latent diffusion-based 3D generation approaches such as TRELLIS and TripoSG.
> These methods performs generation in a high-dimensional latent space, which often leads to hallucinated geometry, shape deviations, and inconsistencies across viewpoints—particularly when multiple input images are involved (see Figure 12 in Supplementary).
> As a result, they struggle to maintain strong pixel-scene and cross-see alignment.
>
> In contrast, we retain the strengths of generative modeling for completion and shift the generative process to a lightweight decoder and leverage a feed-forward 3D foundation model for image-to-latent mapping.
> This significantly constrains the generative space and suppresses stochastic ambiguity.
> Consequently, our method produces geometrically faithful point clouds with visual grounding quality comparable to pixel-aligned approaches, as demonstrated in Figures 6 and 7 (main paper).
>
> > **[W3] NOVA3R’s non-pixel-aligned, point-wise decoding lacks efficient scalability for handling massive point volumes, making it challenging to adapt to tasks requiring extensive multi-view input or high-density point cloud output.**
>
> While pixel-aligned methods can leverage per-ray parallelization tied to image pixels, such parallelism is still constrained at the per-image level (e.g., in VGGT), as points originating from the same image remain coupled in the DPT head.
> More importantly, these methods often need to process duplicated points from multiple views, leading to significant computational overhead when handling large-scale scenes with many input images.
> In contrast, our point-wise decoding naturally uses fewer points (tokens) to represent the scene, making it more scalable to large-scale scenes.
>
> For NOVA3R, a potential direction to enhance scalability with massive point volumes is to exploit submap-based scene token partitioning,
> eg. using the sparse point cloud from COLMAP as guidance.
> Specifically, we can divide the global scene tokens into spatially coherent chunks, each representing a local submap, and decode these submaps independently with fewer points per chunk.
> This enables parallel decoding across chunks and effectively mitigates the memory and computational burden associated with decoding a single large global point cloud.

---

> ### Author Response · Authors · 2025-11-20
> **Official Response by Authors (Part 2)**
>
> > **[Q1] First, compared with end-to-end structures, the two-stage training design requires additional training efforts, and it is worth discussing whether the training complexity can be optimized.**
>
> 1. For comparisons with end-to-end structures, please refer to **[W1]**.
>
> 2. Regarding training complexity, we empirically explored single-stage end-to-end training but found it to be highly unstable and prone to divergence, particularly in the early stages of training. To further reduce complexity, potential directions include using stronger geometric supervision (e.g., optimal transport-based losses) to stabilize training or employing a pre-trained image tokenizer (e.g., DINOv3) with frozen weights to regularize the latent space, training only the flow-matching module.
>
> > **[Q2] Second, the paradigm of "compression-decompression" plus diffusion-based decoding may bring uncertainty to the consistency between the reconstructed point cloud and the input image, and further verification of this consistency is needed.**
>
> In fact, our design specifically addresses and mitigates the uncertainty commonly observed in diffusion-based latent 3D generation.
> As discussed in **[W2]**, unlike methods such as TripoSG and TRELLIS which tend to hallucinate structures and struggle to maintain alignment with the input imagery, our approach employs a feed-forward image-to-latent mapping and a lightweight decoder, which together impose stronger geometric and visual constraints on the generative process.
>
> This is quantitatively validated in Table 1 in the main paper: TripoSG and TRELLIS exhibit notably lower accuracy on visible regions (directly reflecting consistency with input images), while our method achieves performance comparable to pixel-aligned methods like VGGT, despite requiring significantly less training data.
> These results demonstrate that our compression–decoding design not only preserves consistency but also provides reliable, visually grounded reconstructions.
>
> > **[Q3]  Third, the point-wise decoding nature makes it difficult for the model to scale to scenarios with an extremely large number of input images, as the increase in point cloud quantity will lead to significant computational bottlenecks.**
>
> Modeling extremely large-scale scenes with numerous input images is a computational bottleneck for current learning-based 3D reconstruction methods. This challenge is even more severe for pixel-aligned methods such as VGGT and CUT3R, as they must handle duplicated points across multiple views. In contrast, point-wise decoding naturally uses fewer points (tokens) to represent the scene, making it more scalable to large-scale environments. However, the number of points required for high-fidelity reconstruction can vary significantly across different scene scales, which calls for adaptive strategies, such as using sparse point maps from COLMAP to guide point sampling.
>
> For scalability in handling large point clouds during inference, please refer to our response in **[W3]**, where we discuss submap-based parallel decoding to effectively reduce memory and computational overhead. For scenarios with a large number of input images that cannot fit into GPU memory simultaneously, an online image encoder such as CUT3R can be employed. This encoder maintains a fixed-size memory buffer, allowing incremental processing of images without increasing memory usage (also see our reply to **[PSme-W3]**). This strategy enables our framework to scale to long image sequences while preserving efficiency, which we leave for future work.

---

> ### Comment · Reviewer_Fedf · 2025-11-28
> **Comment**
>
> Thanks for your response, and let me provide further clarification regarding our feedback on Question 3. Specifically, while VGGT/CUT3R also encounter challenges in large-scale scenes, they essentially perform computations at the patch level (e.g., each token corresponds to an 8×8 point region). In contrast, the diffusion-based VAE in your framework conducts point-level computations directly. This distinction gives rise to the representational bottleneck—which is my primary concern here. I am concern about that when you raise your input views, if the VAE will be the bottleneck of your whole framework?
>
> Importantly, this is by no means a critique of your paper’s quality; rather, I believe this aspect merits explicit discussion and acknowledgment to ensure it is fully considered in the work.

---

> ### Author Response · Authors · 2025-11-28
> **Official Comment by Authors**
>
> Dear Reviewer Fedf,
>
> Thanks for the clarification and for raising this insightful question regarding the representational bottleneck. This is indeed a valid concern, and we have considered it during the design of our decoder to ensure the AE does not become a bottleneck when scaling up point computations. Our efficient flow-matching decoder achieves a **linear complexity** ($O(N)$) with respect to the large number of inference points ($N$). This is accomplished by replacing direct point self-attention with a compressed, token-based mechanism utilizing a fixed set of $M$ scene tokens ($M=768 \ll N$). Specifically, the computational cost is structured around three steps: cross-attention (Point-to-Token: $O(N M)$), followed by self-attention (Token-to-Token: $O(M^2)$), and finally, cross-attention (Token-to-Point: $O(N M)$). This results in a total complexity of $O(2 M N + M^2)$. As $M$ is fixed and significantly smaller than $N$, the overall cost effectively linear in $N$. This token-based decoder design guarantees scalability, allowing our framework to robustly handle large point cloud inputs (e.g., $N \approx 400\text{k}$) during inference.
>
> For larger scenes, the number of scene tokens may become a bottleneck. A potential solution is to divide the global scene tokens into spatially coherent chunks, with each chunk representing a local submap. As a result, we only need to decode a relatively small amount of points for each local submap.
>
>
> Best regards,
>
> Authors

---

### Official Review · Reviewer_3fs2 · 2025-10-31

**Soundness:** 3
**Presentation:** 3
**Contribution:** 2
**Rating:** 2
**Confidence:** 4

**Summary:**

This paper proposes NOVA3R, a non-pixel-aligned 3D reconstruction method that effectively improves occluded region completion through a scene-token mechanism and flow-matching loss. It provides a feasible solution for the non-pixel-aligned paradigm.

**Strengths:**

As a non-pixel-aligned 3D reconstruction framework, NOVA3R decouples reconstruction from pixel-ray binding via a global Scene Token mechanism. It completes occluded regions, addressing the flaw of traditional pixel-aligned methods (e.g., DUSt3R, VGGT).

**Weaknesses:**

1. Innovation Needs Further Quantification. The core innovation of the paper lies in the synergy between the non-pixel-aligned paradigm and Scene Token. It is recommended to quantify the token’s contribution to global structure modeling.
2. Qualitative Results Require Objective Quantitative Support. The core goal of qualitative experiments (Figures 6, 7) is to verify "occlusion completion" and "physical plausibility," but current evaluations rely solely on visual judgment, leading to potential subjectivity in assessing effectiveness. It is suggested to supplement quantitative metrics (e.g., hole area ratio, point cloud density variance) to objectively measure differences between NOVA3R and baselines. Furthermore, enrich qualitative results for complex scenes to highlight the method’s advantages in challenging scenarios.
3. Validation Scope for Scene Completion Needs Expansion. Verification is only conducted under  (single view) and  (two views), which is insufficient to fully demonstrate the gain of multi-view input for complete reconstruction. It is recommended to add experiments to analyze the relationship between the number of views and reconstruction completeness  as well as physical plausibility.
4. Comparative Experiments Need Clear Task Boundaries and Explanations for Omitted Comparisons. When comparing with pixel-aligned methods (e.g., DUSt3R, VGGT), it is necessary to more explicitly explain the "impact of task differences on metrics" (e.g., VGGT outperforms in CD because it does not need to handle occlusion completion). For the omission of Amodal3R in Object Completion experiments, supplement explanations for the exclusion (e.g., Amodal3R requires mask input and only works for object-level reconstruction). If comparisons are deemed necessary, design adaptive experiments (e.g., providing NOVA3R with the same masks as Amodal3R) before conducting comparisons.
5. Analysis of Flow-Matching Loss Needs Deepening. The paper only mentions using flow-matching loss to address unordered point cloud supervision but fails to analyze its advantages over traditional losses. It also does not explain the tuning strategy for loss hyperparameters (e.g., sampling strategy for time step ). It is recommended to add ablation experiments comparing "different loss functions" to verify the necessity of flow-matching for unordered point cloud encoding. Additionally, explain the tuning differences of the loss across datasets.

**Questions:**

1. Innovation Needs Further Quantification. The core innovation of the paper lies in the synergy between the non-pixel-aligned paradigm and Scene Token. It is recommended to quantify the token’s contribution to global structure modeling.
2. Qualitative Results Require Objective Quantitative Support. The core goal of qualitative experiments (Figures 6, 7) is to verify "occlusion completion" and "physical plausibility," but current evaluations rely solely on visual judgment, leading to potential subjectivity in assessing effectiveness. It is suggested to supplement quantitative metrics (e.g., hole area ratio, point cloud density variance) to objectively measure differences between NOVA3R and baselines. Furthermore, enrich qualitative results for complex scenes to highlight the method’s advantages in challenging scenarios.
3. Validation Scope for Scene Completion Needs Expansion. Verification is only conducted under  (single view) and  (two views), which is insufficient to fully demonstrate the gain of multi-view input for complete reconstruction. It is recommended to add experiments to analyze the relationship between the number of views and reconstruction completeness  as well as physical plausibility.
4. Comparative Experiments Need Clear Task Boundaries and Explanations for Omitted Comparisons. When comparing with pixel-aligned methods (e.g., DUSt3R, VGGT), it is necessary to more explicitly explain the "impact of task differences on metrics" (e.g., VGGT outperforms in CD because it does not need to handle occlusion completion). For the omission of Amodal3R in Object Completion experiments, supplement explanations for the exclusion (e.g., Amodal3R requires mask input and only works for object-level reconstruction). If comparisons are deemed necessary, design adaptive experiments (e.g., providing NOVA3R with the same masks as Amodal3R) before conducting comparisons.
5. Analysis of Flow-Matching Loss Needs Deepening. The paper only mentions using flow-matching loss to address unordered point cloud supervision but fails to analyze its advantages over traditional losses. It also does not explain the tuning strategy for loss hyperparameters (e.g., sampling strategy for time step ). It is recommended to add ablation experiments comparing "different loss functions" to verify the necessity of flow-matching for unordered point cloud encoding. Additionally, explain the tuning differences of the loss across datasets.

---

> ### Author Response · Authors · 2025-11-20
> **Official Response by Authors (Part 1)**
>
> We sincerely thank the reviewer for the insightful comment and constructive feedback. We have carefully addressed each comment and hope that our responses clarify the concerns. We would be glad to further elaborate if any additional questions remain.
>
> > **[Q1] Innovation Needs Further Quantification. It is recommended to quantify the token’s contribution to global structure modeling.**
>
> We indeed have quantified the contribution of Scene Tokens for global structure modeling through ablation studies in Table 5 (main paper),
> and discuss details in Sec. 4.4. Specifically, we evaluate:
> (1) Init Query tokens, which determine how the scene tokens are initialized, and
> (2) the Number of Tokens, which reflects the capacity of the global scene representation.
> As shown in Table 5, hybrid query initialization yields the best reconstruction accuracy, confirming its importance for effective global reasoning. Furthermore, increasing the number of Scene Tokens consistently improves reconstruction completeness and structural consistency, highlighting their role in modeling global context, although this naturally comes with increased computational cost.
>
> Except for the Scene Token design,
> our key contribution is the formulation of a **unified non-pixel-aligned reconstruction pipeline** with minimal assumptions, applicable to both object-level and scene-level completed reconstruction tasks. This non-pixel-aligned paradigm addresses key limitations of the popular pixel-aligned methods (e.g., DUST3R, VGGT), which often suffer from **incomplete point clouds, duplicated geometry, and 3D inconsistencies in overlapping regions**. Furthermore, our integration of a feed-forward transformer encoder with a lightweight flow-matching decoder is another key innovation of our method, effectively bridging pixel-aligned reconstruction and latent 3D generation.
>
> > **[Q2]  It is suggested to supplement quantitative metrics (e.g., hole area ratio, point cloud density variance) to objectively measure differences between NOVA3R and baselines. Furthermore, enrich qualitative results for complex scenes to highlight the method’s advantages in challenging scenarios.**
>
> Thank you for your suggestion. We have supplemented quantitative evaluations using hole area ratio and point cloud density variance on the SCRREAM dataset. Specifically, the hole area ratio is measured by checking, for each ground-truth point, whether a predicted point lies within a distance threshold of 0.1. The density variance is computed by first estimating local point density using k-nearest neighbors (k=16), followed by computing the global variance across all points. As shown in Table R3, our method consistently achieves a significantly lower hole ratio and density variance across different numbers of input views, indicating more complete and uniformly distributed reconstructions.
>
> To further highlight advantages in challenging scenarios, we provide additional qualitative results for complex indoor scenes in Figure 11 (Supplementary), and new results for large-scale outdoor scenes in Figure 13 (Supplementary). These examples clearly demonstrate our method’s ability to recover fine structures, fill self-occluded regions, and maintain spatial consistency in large and cluttered environments. We have included these results in the revision.
>
>
> | Method |   Complete (K=1)   |                 |   Complete (K=2)   |                 |   Complete (K=4)   |                 |
> |--------|--------------------|------------------|--------------------|------------------|--------------------|------------------|
> |        | Hole Ratio ↓       | Density Var ↓    | Hole Ratio ↓       | Density Var ↓    | Hole Ratio ↓       | Density Var ↓    |
> | DUST3R | 0.317              | 7.758            | 0.237              | 6.553            | 0.257              | 4.801            |
> | CUT3R  | 0.363              | 8.402            | 0.237              | 6.554            | 0.326              | 4.658            |
> | VGGT   | 0.307              | 7.105            | 0.238              | 6.546            | 0.261              | 5.217            |
> | **Ours** | **0.088**        | **5.127**        | **0.121**          | **2.188**        | **0.134**          | **1.881**        |
>
> *Table R3: Hole ratio and point cloud density variance on SCRREAM.*
>
> *Note that under our definition of complete reconstruction, the hole ratio measures how well the geometry is recovered within the input-view frustum. As the number of input views increases, the frustum coverage becomes larger, making it more challenging to maintain a low hole ratio.*

---

> ### Author Response · Authors · 2025-11-20
> **Official Response by Authors (Part 2)**
>
> > **[Q3]  Verification is only conducted under (single view) and (two views), which is insufficient to fully demonstrate the gain of multi-view input for complete reconstruction. It is recommended to add experiments to analyze the relationship between the number of views and reconstruction completeness as well as physical plausibility.**
>
>
> The scene-level geometry is generally unbounded, making a unified definition of global completeness inherently ambiguous.
> Hence,
> our definition of complete reconstruction is to recover all geometry within the frustum of the input views,
> as described in Sec. 3.1 and illustrated in Figure 4 (main paper).
> Under this definition,
> the **single-view setting is indeed the most challenging**,
> and we therefore include it as a core part of our main evaluation.
>
> Moreover, we would like to clarify that our experiments are not limited to single-view and two-view inputs; results with four input views are already included in Sec. 4.2 (Figure 7, main paper) and further shown in Figure 11 (Supplementary).
> Although trained only with two views, our model generalizes well to four-view inputs, leading to improved completeness and physical plausibility. As shown in Table R3 (see **[Q2]**), the density variance consistently decreases from one to four views, indicating better physical plausibility with more views. Direct comparison of completeness across different view counts is non-trivial due to varying input-image combinations filtered by overlap constraints. Nevertheless, even in unseen four-view cases, our method still outperforms pixel-aligned baselines in both hole ratio and density variance.
>
> > **[Q4-1] When comparing with pixel-aligned methods (e.g., DUSt3R, VGGT), it is necessary to more explicitly explain the ``impact of task differences on metrics".**
>
> In our comparison with pixel-aligned methods (DUST3R, VGGT), we have clearly delineated two distinct evaluation settings in Table 1 (main paper): the visible setting and the complete setting.
> For visible setting,
> we follow the same evaluation protocol as DUST3R and VGGT, where the ground truth contains only the visible points from the input views.
> Under this setting, our method achieves comparable accuracy,
> even though it is trained on a much smaller-scale training dataset than other baselines.
> For complete setting,
> we use the full point cloud as ground truth, including occluded and unseen regions.
> Under this setting, our method clearly outperforms DUST3R and VGGT,
> as pixel-aligned methods are inherently unable to recover unseen geometry.
> This demonstrates the effectiveness of our non-pixel-aligned paradigm for scene-level completion.
>
> > **[Q4-2] For the omission of Amodal3R in Object Completion experiments, supplement explanations for the exclusion.**
>
> Our main focus is *scene-level* completion,
> which contains more complex occlusion patterns and relationships among multiple objects and background,
> while the *object-level* experiments mainly demonstrate the generalization ability of our method.
> However,
> we would like to clarify why Amodal3R is not included in our *object-level* completion experiments.
> 1. Amodal3R assumes the presence of explicit occluder (in front of the target object) masks as input, which is not available in our setting.
> 2. Amodal3R is designed for object-centric scenarios and relies on mask guidance, while our method focuses on *self-occlusion* completion at both object and scene levels without requiring any external masks or semantic inputs.
> 3. To ensure fairness, we have compared with TRELLIS, the backbone of Amodal3R which does not require mask input, making it more aligned with our problem setting.

---

> ### Author Response · Authors · 2025-11-20
> **Official Response by Authors (Part 3)**
>
> > **[Q5]  It is recommended to add ablation experiments comparing "different loss functions" to verify the necessity of flow-matching for unordered point cloud encoding. Additionally, explain the tuning differences of the loss across datasets.**
>
> To verify the necessity of flow-matching for unordered point cloud encoding, we conducted an ablation using the same architecture but replaced the flow-matching loss with the classical Chamfer Distance loss. Both models were trained on SCRREAM (Stage 1) under the same settings. As shown in Table R4, flow-matching yields significantly better reconstruction quality and generalization. This confirms that Chamfer Distance struggles in scene-level cases due to its nearest-neighbor formulation, which is computationally expensive, sensitive to density imbalance, and unable to model global structure. In contrast, flow-matching loss learns a continuous distribution and produces stable, complete, and globally consistent reconstructions.
>
> Regarding loss tuning, we use the standard flow-matching formulation with a cosine noise scheduler during training, uniformly sampling timesteps from [0,1], and use a fixed step size of 0.04 during inference.
> Importantly, we apply the **exact same loss configuration** for both object-level and scene-level datasets, without additional tuning, demonstrating the robustness and general applicability of the flow-matching objective in our pipeline.
>
>
> | Training Loss     | CD ↓     | FS@0.1 ↑ | FS@0.05 ↑ | FS@0.02 ↑ |
> |-------------------|----------|----------|-----------|-----------|
> | Chamfer Distance  | 0.024    | 0.981    | 0.907     | 0.575     |
> | **Flow-Matching** | **0.011** | **0.999** | **0.993** | **0.904** |
>
>
> *Table R4: Ablations on different training loss functions.*

---

> > ### Comment · Reviewer_3fs2 · 2025-11-28
> > **Comment**
> >
> > Thanks for the detailed responses that address most of my concerns. I would like to raise an issue that likely impacts the significance of the paper’s contributions: the paper’s claimed physical reasonableness is limited to resolving duplicated points in overlapping regions. However, such redundant point artifacts are barely a critical bottleneck for achieving true physical plausibility. They can be easily alleviated via simple post-processing steps (e.g., voxel-grid filtering, density thresholding), obviating the need for a fundamental paradigm shift to a non-pixel-aligned formulation.

---

> ### Author Response · Authors · 2025-11-28
> **Official Comment by Authors**
>
> Dear Reviewer 3fs2,
>
> Thanks for raising this insightful question regarding physical plausibility. We agree with the reviewer that if the only issue were redundant points (high density), simple post-processing would be sufficient. However, the "physical plausibility" we claim addresses a deeper challenge than just density: **3D geometric consistency in overlapping regions**, which simple filtering cannot resolve. Beyond redundant points, pixel-aligned methods face other limitations:
> 1. **Multi-layer Artifacts**. Pixel-aligned methods (e.g., VGGT, CUT3R) often suffer from 3D inconsistencies across multiple views due to prediction noise. For example, a single wall viewed from multiple angles may be reconstructed as multiple distinct surfaces at slightly different positions. The problem becomes worse when the number of input images increases. Post-processing filters would treat these as valid, separate geometries rather than merging them. In comparison, our non-pixel-aligned formulation inherently aggregates these features during generation, avoiding the creation of disjoint surfaces in the first place.
> 2. **Depth-Dependent Density Issues**. In pixel-aligned reconstruction, the density of the point cloud directly depends on the distance from the camera center, causing point density to degrade significantly at larger depths. Consequently, a single density threshold for fusion is ineffective: a threshold strict enough to clean the near-field will wipe out valid geometry in the far-field. Our non-pixel-aligned formulation alleviates this by applying Farthest Point Sampling (FPS) on targets during training, which promotes uniform coverage.
>
> Best regards,
>
> Authors

---

### Official Review · Reviewer_PSme · 2025-10-31

**Soundness:** 3
**Presentation:** 3
**Contribution:** 3
**Rating:** 8
**Confidence:** 4

**Summary:**

This manuscript presents NOVA3R, a non-pixel-aligned visual transformer for amodal 3D reconstruction from unposed images. Unlike prior methods tied to pixel-level supervision, NOVA3R decouples geometry prediction from image pixels by learning a global scene representation. The approach leverages a diffusion-based 3D autoencoder trained with a flow matching loss, and introduces learnable scene tokens to aggregate multi-view information. The method demonstrates state-of-the-art performance across object-level and scene-level benchmarks, outperforming both single- and multi-view baselines.

**Strengths:**

- The paper is well-written and well-presented. The provided figures are clear and informative. The accompanying video shows clearly the method in action.
- The idea of predicting non-pixel-aligned geometry is novel and interesting. While the literature of 3D generation is rich, framing the tasks as an intersection of DUST3R-style prediction and 3D-native prediction is clever and refreshing.
- The method achieves state-of-the-art performance both on the SCRREAM dataset and NRGBD/7-Scenes dataset, demonstrating the effectiveness of the proposed method.

**Weaknesses:**

- The runtime performance of the flow-matching decoder is not analyzed. It would be nice to show how the proposed method compares to a regular point cloud decoder without the diffusion process.
- It is unclear how image-based properties, such as camera poses or intrinsics, or depth maps, can be derived. While the entire scene is put under the coordinate system of the first view, it is not clear how accurate it is for other views. Pose or depth accuracy evaluation is not provided.
- It is not clear how the proposed method can be extended to more than 4 views, i.e., it is not clear how a global bundle adjustment problem can be formulated as in DUST3R to incorporate videos of arbitrary length.

**Questions:**

- While the method focuses on amodal 3D reconstruction, do you have any thoughts on how image-based properties such as camera poses or intrinsics, or depth maps can be derived from the proposed scene representation?
- How does the method generalize to more views that covers larger areas of the scene? e.g. how the method performs on autonomous driving datasets such as nuScenes or Waymo?
- What is the maximum pose gap between the two input images? How large the portion of the scene can be completed?
- How can the method be extended to handle dynamic scenes? How can the method handle incremental reconstruction?

---

> ### Author Response · Authors · 2025-11-20
> **Official Response by Authors (Part 1)**
>
> We sincerely thank the reviewer for the insightful comment and constructive feedback. We have carefully addressed each comment and hope that our responses clarify the concerns. We would be glad to further elaborate if any additional questions remain.
>
> > **[W1] It would be nice to show how the proposed method compares to a regular point cloud decoder without the diffusion process.**
>
> We reported the runtime performance comparison between different decoders:
> (1) our flow-matching decoder,
> (2) a regular global point cloud decoder trained using Chamfer Distance, and
> (3) a mesh-based decoder widely used in 3D object generation from TripoSG.
> The results are shown in below.
> For the flow-matching decoder, we use a sample size of 50,000 and a time step of 0.04 during inference. All experiments are conducted on an NVIDIA GeForce RTX 4090 24 GB GPU.
> As shown in Table R1, our method is slower than the single-step point cloud decoder due to the iterative nature of flow matching.
> However, it is significantly faster than mesh-based decoders, which require implicit surface evaluation and complex geometric processing.
>
>
> | Decoder | Single-step Decoder | Flow-Matching Decoder | Mesh Decoder (TripoSG) |
> |---|---:|---:|---:|
> | Runtime | 0.557 s | 2.985 s | 9.551 s |
>
> *Table R1: Comparison of decoder runtime.*
>
>
> Although more efficient, the chamfer-based decoder cannot reliably support scene-level, non-pixel-aligned completion task (also see reply to **[3fs2-Q5]**). A chamfer-distance-based decoder typically requires a fixed number of output points, making it difficult to generalize across varying scene scales and different numbers of input views.
> Additionally, its local nearest-neighbor matching often fails to capture the global structure, resulting in uneven point distributions and missing details (see Table R2).
> In contrast,
> - Our flow-matching decoder can flexibly generate varying numbers of points by adjusting the denoising sample size.
> - It models global geometry and effectively handles variable scene scales and coverage.
> - The implementation uses standard flow matching and can be further accelerated with techniques such as consistency distillation or truncated flow steps.
>
>
> | Training Loss      | CD ↓   | FS@0.1 ↑ | FS@0.05 ↑ | FS@0.02 ↑ |
> |--------------------|-------:|---------:|----------:|----------:|
> | Chamfer distance   | 0.024  | 0.981    | 0.907     | 0.575     |
> | Flow-matching      | **0.011** | **0.999** | **0.993**  | **0.904**  |
>
> *Table R2: Ablations on different training loss functions.*
>
> > **[W2] It is unclear how image-based properties, such as camera poses or intrinsics, or depth maps, can be derived. While the entire scene is put under the coordinate system of the first view, it is not clear how accurate it is for other views.**
>
> Our main focus is on global reconstruction with physical plausibility (with less duplicated and inconsistent geometry in the overlapping regions) and completed geometry.
> While the camera poses, or intrinsic, or depth maps can be derived by using additional DPT heads for each input image token, similar to VGGT,
> and the global scene tokens may provide global context for better prediction,
> this goes beyond the scope of this paper.
> We leave this as a future direction.
>
> > **[W3] It is not clear how the proposed method can be extended to more than 4 views, i.e., it is not clear how a global bundle adjustment problem can be formulated as in DUST3R to incorporate videos of arbitrary length.**
>
> Similar to VGGT and MapAnything,
> our method is inherently view-agnostic and can be directly extended to multiple input views (e.g., 16 or more) with the global attention mechanism.
> Our method imposes no limit on the number of views, and the only practical constraint is GPU memory.
> For longer sequences or full videos that exceed GPU memory, streaming strategies (e.g., StreamVGGT)
> or recurrent encoders (e.g., CUT3R) can be adopted to incrementally fuse input views while maintaining a fixed-size memory, enabling global scene reasoning over arbitrary-length input.

---

> ### Author Response · Authors · 2025-11-20
> **Official Response by Authors (Part 2)**
>
> > **[Q1] While the method focuses on amodal 3D reconstruction, do you have any thoughts on how image-based properties such as camera poses or intrinsics, or depth maps can be derived from the proposed scene representation?**
>
> Please refer to **[W2]**.
>
> > **[Q2] How does the method generalize to more views that covers larger areas of the scene? e.g. how the method performs on autonomous driving datasets such as nuScenes or Waymo?**
> 1. For more views, please refer to **[W3]**.
>
> 2. For outdoor scenarios such as autonomous driving datasets, our method can be directly applied, as it is designed for general 3D scene reconstruction and completion. To evaluate its capability in such environments, we finetuned our model on Virtual KITTI 2 and found that it successfully reconstructs outdoor scenes. As shown in Figure 13 (Supplementary), NOVA3R produces high-quality reconstructions in outdoor settings, further demonstrating its ability to handle both indoor and outdoor scenarios.
>
> > **[Q3] What is the maximum pose gap between the two input images? How large the portion of the scene can be completed?**
>
> 1. In our testing dataset,
> the maximum pose gap between two input images is 30\% overlappling region measured in the point cloud covisibility.
> Our method is able to reliably map these views and produce consistent scene reconstructions.
> Since the model is trained on data with a wider range of viewpoint variations, it can potentially generalize to even larger pose gaps beyond those observed in the evaluation set.
>
> 2. Regarding scene completion, the hole area ratio on the evaluation dataset ranges from 5.3\%  to 48.6\%, depending on input coverage and viewpoint diversity.
> Within this range, our method can effectively recover large continuous regions and infer missing structures, demonstrating robustness even under sparse view overlap.
>
> > **[Q4] How can the method be extended to handle dynamic scenes? How can the method handle incremental reconstruction?**
>
>
> 1. Following works such as Dynamic Point Maps, ST4RTrack, and Stereo4D, our framework can be naturally extended to 4D scene representation by conditioning the latent geometry on time or a target frame. Unlike these methods, which reconstruct separate per-frame point clouds and track them over time, our approach has the potential to represent the entire dynamic scene in a unified canonical coordinate system (e.g., the first frame), which helps avoid temporal drift and improves structural consistency. To fully support dynamic reconstruction, static geometry and motion would need to be decoupled by introducing a motion decoder that predicts per-point motion vectors. This extension is beyond the scope of the current paper but will be explored in our future work.
>
> 2. For incremental reconstruction, please refer to **[W3]**.

---

### Author Response · Authors · 2025-11-28
**General Response**

We sincerely thank the reviewers for their insightful comments, valuable feedback, and recognition of the strengths of our work:
- **Important Formulation**: Reviewers acknowledged our motivation and the importance of our proposed non-pixel-aligned formulation (PSme, 3fs2, Fedf, t2bA), which directly addresses the incomplete, duplicated, and inconsistent geometry issues inherent in pixel-aligned methods (Fedf, t2bA).
- **Novelty and Contributions**: Reviewers found our integration of generative 3D concepts into 3D-native prediction refreshing (PSme), novel (t2bA), and highly valuable (Fedf). They praised the effectiveness of the flow-matching decoder for learning unordered point clouds (t2bA, Fedf) and the global scene tokens for view-agnostic representation (Fedf).
- **Comprehensive Experiments**: Reviewers found the experiments well-validated across scene-level and object-level datasets, achieving state-of-the-art performance (PSme, t2bA).

---

We have updated the manuscript according to the reviewers’ suggestions. All revisions in the updated version are highlighted in **red**:

1. **Additional experiments**:
    - Add quantitative evaluation of hole area ratio and point cloud density variance (3fs2-Q2; L.409).
    - Add experiments with four-view inputs and analyze the relationship between the number of views, completeness, and physical plausibility (3fs2-Q3; L.414).
    - Add an ablation study comparing flow-matching loss and Chamfer Distance loss in both reconstruction quality and runtime (PSme-W1, 3fs2-Q5; L.513).
    - Add uncertainty verification to compare reconstruction consistency with latent 3D generation approaches (Fedf-W2/Q2; Appendix A.4).
    - Add experimental results on outdoor and autonomous driving scenes (PSme-Q2, 3fs2-Q2, t2bA-Q2; Appendix A.5).

2. **Additional discussions and details**:
    - Add clearer explanation of contributions and quantify the impact of scene tokens (3fs2-Q1; L.101, L.468).
    - Add clarification of the differences between visible and complete reconstruction settings (3fs2-Q4; L.377).
    - Add details on flow-matching loss configurations and the two-stage training scheme (3fs2-Q5, Fedf-Q1; L.322, Appendix A.1).
    - Add discussion on extensions to large-scale scenes and dynamic scenes (PSme-Q4, Fedf-Q3, t2bA-W3; Appendix A.6).


We look forward to engaging discussions and appreciate the opportunity to refine our work based on the reviewers' constructive feedback.

---

### Meta-Review · Area_Chair_a6ZY · 2026-01-06

**Summary:**

This paper proposes a non-pixel-aligned feedforward 3D reconstruction framework that addresses a well-known limitation of pixel-aligned methods, namely the presence of holes and duplicated structures caused by per-ray predictions. By leveraging a global latent autoencoding mechanism inspired by recent latent 3D generation models and integrating it with a feedforward visual transformer, the method produces more complete and physically plausible point clouds from unposed images. While the core idea is conceptually simple and builds on existing autoencoding techniques, the approach is clearly formulated, and the empirical results consistently demonstrate improvements in completeness and geometry quality over strong pixel-aligned baselines. Overall, the paper makes a solid and timely contribution to feedforward 3D reconstruction.

**Reviewer Concerns:**

Concerns addressed or largely mitigated:
Reviewers generally agreed that the paper clearly identifies a real limitation of pixel-aligned reconstruction methods and proposes a reasonable solution. The experimental section is thorough, covering both scene-level and object-level reconstruction, single-view and multi-view settings, and multiple datasets. Reviewers appreciated the quantitative analysis on hole ratios and point density variance, which directly supports the motivation for non-pixel-aligned reconstruction. Ablation studies further validate key design choices, such as the latent autoencoder, scene tokens, and flow-matching decoder.

Existing concerns:
Several reviewers noted that the main idea, introducing a global latent representation to avoid pixel alignment, is relatively straightforward and closely related to recent latent 3D autoencoding works. As a result, the conceptual novelty is somewhat limited. In addition, some claims in the paper could be better calibrated to reflect the incremental nature of the contribution, particularly when positioning the method relative to prior work on latent 3D generation and feedforward reconstruction. These concerns primarily affect framing rather than technical soundness and can be addressed by slightly tempering the claims and clarifying the relationship to existing approaches.

Besides, the extendability of the paper remains questionable. As the method requires complete scenes for training, it can only be trained on synthetic datasets, limiting model scalability. Moreover, the experiments in the paper mainly focus on 1 or 2 view reconstruction, while solving the limited capacity of a fixed number of scene tokens under dense view inputs is a real problem. In general, I believe this paper opens an interesting direction, but many problems are unsolved.

**Reviewer Scores:**

Reviewer PSme: Likely to maintain their original score, as they found the method effective and well supported by experiments, with only minor concerns about novelty.

Reviewer 3fs2: Likely to maintain or slightly increase their score after discussion, given the strong empirical results and clear improvements over pixel-aligned baselines.

Reviewer Fedf: Likely to maintain their score. While they viewed the idea as simple, they acknowledged that the results are solid and the evaluation is comprehensive.

Reviewer t2bA: Likely to maintain their original score, considering the work a practical and well-executed contribution with reasonable scope.

---

### Decision · Program_Chairs · 2026-01-26

Accept (Poster)